

# The Luangwa Rift Active Fault Database and fault reactivation along the southwestern branch of the East African Rift

Luke N. J. Wedmore[1], Tess Turner[1], Juliet Biggs[1], Jack N. Williams[1,2,3], Henry M. Sinchingabula[4], Christine Kabumbu[4], and Kawawa Banda[5]

[1]School of Earth Sciences, University of Bristol, Bristol, UK
[2]School of Earth and Environmental Sciences, Cardiff University, Cardiff, UK
[3]Department of Geology, University of Otago, Dunedin, New Zealand
[4]Department of Geography and Environmental Studies, University of Zambia, Lusaka, Zambia
[5]Department of Geology, School of Mines, University of Zambia, Lusaka, Zambia

**Correspondence:** Luke Wedmore (luke.wedmore@bristol.ac.uk)

**Abstract.** Seismic hazard assessment in slow straining regions is challenging because earthquake catalogues only record events from approximately the last 100 years, whereas earthquake recurrence times on individual faults can exceed 1,000 years. Systematic mapping of active faults allows fault sources to be used within probabilistic seismic hazard assessment, which overcomes the problems of short-term earthquake records. We use Shuttle Radar Topography Mission (SRTM) data to analyse

surface deformation in the Luangwa Rift in Zambia and develop the Luangwa Rift Active Fault Database (LRAFD). The LRAFD is an open-source geospatial database containing active fault traces and their attributes and is freely available at: https://doi.org/10.5281/zenodo.6513691. We identified 18 faults that display evidence for Quaternary activity and empirical relationships suggest that these faults could cause earthquakes up to Mw 8.1, which would exceed the magnitude of historically recorded events in southern Africa. On the four most prominent faults, the median height of Late Quaternary fault scarps varies

between $12.9 \pm 0.4$ and $19.2 \pm 0.9$ m, which suggests they were formed by multiple earthquakes. Deformation is focused on the edges of the Luangwa Rift: the most prominent Late Quaternary fault scarps occur along the 207 km long Chipola and 142 km long Molaza faults, which are the rift border faults and the longest faults in the region. We associate the scarp on the Molaza Fault with possible surface ruptures from two 20th Century earthquakes. Thus, the LRAFD reveals new insights into active faulting in southern Africa and presents a framework for evaluating future seismic hazard.

**1 Introduction**

Earthquakes occur on active faults, and thus the systematic mapping of active faults is a major aim of seismic hazard research (Christophersen et al., 2015; Morell et al., 2020; Styron and Pagani, 2020; Williams et al., 2021, 2022). Within continental rifts, earthquakes on normal faults typically lead to high levels of shaking in their hanging wall basins, which are geomorphically suitable for human habitation and settlement (Bailey et al., 2000; Abrahamson et al., 2008). Consequently, normal faults

inherently create conditions that lead to high seismic risk. Despite the seismic hazards associated with active continental rifting, many active extensional regions around the world still lack systematic maps of active faults. This is particularly a

 

problem along many segments of the East African Rift, where there is a history of infrequent large magnitude earthquakes (Ambraseys and Adams, 1991; Meghraoui, 2016), but the location and activity rates of active faults is poorly known (Skobelev et al., 2004), and population growth over the past 20 years has been rapid (Gerland et al., 2014).

The creation of an active fault database involves defining a criteria to distinguish active faults, systematically map all known faults that fit this criteria, and then collating their geomorphic attributes into a geospatial database (Styron and Pagani, 2020; Styron et al., 2020; Faure Walker et al., 2021; Williams et al., 2021, 2022). The use of active fault databases for fault-source seismic hazard is important in regions such as southern and eastern Africa, where the instrumental and historical records of earthquakes are short compared to the long recurrence times between earthquakes on individual faults (Hodge et al., 2015;

Williams et al.). In recent years, the first active fault databases along the East African Rift System have been developed, using the Malawi Rift as a case study (Williams et al., 2021, 2022), but this has not yet been extended to other rift segments. In this paper, we map the active faults in one poorly studied rift segment, the Luangwa Rift in Zambia. Although it had been thought that the Luangwa Rift is inactive (Banks et al., 1995; Matende et al., 2021; Sun et al., 2021), recent plate modelling (Wedmore et al., 2021) and the evidence of Quaternary activity on the Chipola Fault (Daly et al., 2020) confirm this is an active rift system

(Figure 1). However, until now there has been no systematic map of active faults in the region.

Identifying active faults in a region can reveal new insights into the seismotectonics of a region of active deformation. In southern Africa, this is important as there is debate over 1) the potential magnitude of future earthquake events given that faults may rupture completely or in segments (Jackson and Blenkinsop, 1997; Hodge et al., 2018) – and 2) why the continent is rifting given that tectonic forces are not thought to be sufficient to overcome the strength of the lithosphere (Kendall and

Lithgow-Bertelloni, 2016; Rajaonarison et al., 2021). Mapping faults, and the way in which they are segmented, is vital to addressing these debates. Firstly, the distribution of faults at the surface of the Earth can reveal the strength of the underlying lithosphere (Buck, 1991; Brun, 1999), and secondly fault segment boundaries may act as barriers to earthquake rupture (Aki, 1984; DuRoss et al., 2016). Thus, we develop the Luangwa Rift Active Fault Database (LRAFD), following the framework of the Global Active Fault Database (Styron and Pagani, 2020), and the Malawi Active Fault Database (Williams et al., 2022).

We use the Shuttle Radar Topography Mission (SRTM; Farr et al. (2007)) digital elevation model (DEM) alongside geological maps and previously published analyses to study the tectonic geomorphology of the Luangwa Rift. Based on the discovery of steep fault scarps that offset Quaternary fluvial and alluvial sediments, and incised river valleys, we identify 18 active faults. We then estimate the seismic source properties of these faults using empirical scaling laws (Leonard, 2010). We use the high resolution geomorphology of the fault scarps to identify evidence for fault segmentation and/or multiple earthquakes (Hodge

et al., 2019, 2020). The LRAFD is fully open source and thus available for researchers and practitioners to implement within future regional fault databases and probabilistic seismic hazard analyses (PSHA). By using remote sensing to identify active faults, the outcomes of this research provide targets for future ground-based studies of active tectonics in the Luangwa Rift.





## 2 Tectonic and Geologic Background

The southwestern branch of the East African Rift System (EARS) bifurcates from the western branch of the EARS in Tanzania
and runs through Zambia, Botswana and into Namibia (Figure 1; Fairhead and Girdler, 1969; Reeves, 1972; Scholz et al.,
1976; Fairhead and Henderson, 1977; Daly et al., 2020; Wedmore et al., 2021). The Luangwa Rift is situated in northeastern
Zambia at the northern end of the southwestern branch of the EARS (Figure 1 & 2), and forms the eastern margin of the
Central African Plateau (Daly et al., 2020). It remains unclear whether the onset of rifting along the southwestern branch of
the EARS is contemporaneous with the Oligocene initiation of rifting along the western branch of the EARS (∼25Ma Roberts
et al., 2012). Apatite fission track thermochronometry data from the southwestern branch suggest a period of regional cooling
between 38-22 Ma (Daly et al., 2020). However, Daly et al. (2020) suggest that rifting along the southwestern branch initiated
in the Pliocene (5-3 Ma) at the same time as a period of regional uplift that formed the Central African Plateau.

The Luangwa Rift was active during the Permian-Jurassic breakup of Gondwana (Daly et al., 1989; Banks et al., 1995;
Matende et al., 2021) and during the Cretaceous (Daly et al., 2020). Up to 8,000 m of Permo-Traissic (ie Karoo period in
southern Africa) mainly clastic sediments are unconformably capped by finer grained post-Karoo deposits (Banks et al., 1995).
Although now an amagmatic rift, the Karoo phase of rifting was concomitant with the emplacement of diamond bearing
lamproites, suggesting that this was possibly a rare example of rifting of thick (180-200 km), cold (≤42 mW/m2) cratonic
lithosphere (Ngwenya and Tappe, 2021). The basin also experienced folding during the Late Jurassic-Early Cretaceous (Banks
et al., 1995). The post-Karoo deposits are up to  500 m thick at the northern end of the Rift, and in the southern part of the
Rift, the upper- and post-Karoo sediments are indistinguishable in seismic reflection data (Banks et al., 1995). Negative Vs
anomalies observed in the top few km beneath the southern Luangwa Rift are suggestive of loose sediments (Wang et al.,
2019), and are similar to the low Vs anomalies observed beneath the Malawi Rift, which has up to  5.5 km of syn-rift sediments
from the current post-Miocene phase of rifting in East Africa (Wang et al., 2019; Scholz et al., 2020).

The Luangwa Rift is  130 km wide and  500 km long, with two main escarpments that are greater than  1 km high (Figure
2). The orientation of the rift follows the surface trace of the Mwembeshi shear zone (also referred to as the Mwembeshi
Dislocation Zone; de Swardt et al., 1965), a lithospheric scale structure that may have reactivated along a suture between
the Irumide and Southern Irumide orogenic belts and which accommodated ENE-WSW dextral displacement during the late
Proterozoic (Daly et al., 1989; Sarafian et al., 2018; Alessio et al., 2019). Little is known about the lithology of the Mwembeshi
shear zone as it is largely obscured by the sediments in the Luangwa Rift, other than that it displays a NE trending magnetic
fabric, and contains eclogite (i.e. mafic) intrusives (Vrána et al., 1975; Sarafian et al., 2018). The Nyamadzi shear zone is a
splay of the Mwembeshi shear zone and is comprised of planar, vertically dipping fabrics within a wide variety of lithologies
including ultramylonitic granites, highly deformed quartzites and mafic igneous gabbros and amphibolites (Johnson et al.,
2006).

Daly et al. (2020) found evidence that shows that the Luangwa Rift has been active during the Quaternary, whereas others
suggest that rifting ceased in the Mesozoic (Banks et al., 1995; Matende et al., 2021; Sun et al., 2021). The rift is hosted in
150-160 km thick lithosphere (Priestley et al., 2018), with a crustal thickness of 41-45 km (Sun et al., 2021), and seismicity





occurs down to 29 km (Craig et al., 2011; Craig and Jackson, 2021). Tectonic plate modelling of southern Africa suggests that the Luangwa Rift accommodates $0.7 \pm 0.3$ mm/yr of extension between the San and Nubian plates along an azimuth of 108°(Wedmore et al., 2021), and historical earthquake data shows Mw 6.7 and Mw 6.5 earthquake events occurring in 1919 and

1940 (International Seismological Centre, 2021 and NEIC). However, there has been no prior systematic attempt to document the active faults within the Luangwa Rift and characterize their geomorphic attributes.

## 3    Methods

### 3.1    Compiling the Luangwa Rift Active Fault Database (LRAFD)

#### 3.1.1    Mapping Faults

We use a 30 m (1-arc second) Shuttle Radar Topography Mission (SRTM; Farr et al. (2007)) digital elevation model (DEM) to map the active faults in the Luangwa Rift, which has an absolute height error (90%) of 5.6 m in Africa (Rodriguez et al., 2005). SRTM data has been successfully used for remote investigation and mapping of active faults in southern Africa (Kinabo et al., 2007; McCarthy, 2013; Laõ-Dávila et al., 2015), and is available for free with global coverage. We georeferenced a 1:1,000,000 scale geological map of Zambia (Priday and Camps, 1960), and used this alongside previous academic publications and the

topographic data to identify active faults. We combined these resources with Google Earth imagery to correlate each fault in a range of different datasets.

Active faults in the LRAFD are defined as having a high likelihood of producing significant seismicity under the current tectonics regime (Styron and Pagani, 2020). We make this assessment based on whether the faults display evidence of offsetting Quaternary sediments in the Luangwa basin as it is not clear how long the current tectonic regime along the southwestern

branch of the EAR has been active (see discussion above). Although Quaternary sedimentation in the Luangwa Rift is minimal compared to Karoo sediments (Dixey, 1937; Utting, 1988; Bishop et al., 2016), exploratory petroleum cores and cosmogenic dates from archaeological surveys identified 40 m of sediment that is Quaternary aged (Barham et al., 2011), which is of comparable thickness to the juvenile southern Malawi Rift (Wedmore et al., 2020b). We identified steep scarps that offset these Quaternary sediments, which demonstrates evidence of recent fault activity. Although Daly et al. (2020) suggest that these

steep scarps are <10 ka in age, this is not based on any definitive geochronology, so we prefer the term 'Quaternary' for the age of these scarps.

Active faults in the LRAFD are a defined as having high likelihood of producing significant seismicity under the current tectonic regime (Styron and Pagani, 2020). We use the following criteria as indicators of active faulting: i) prominent, steep (20-30°) linear scarps at the base of the footwall escarpments offsetting Quaternary sediments; ii) evidence of footwall uplift

in river channels such as river gradient steepening, channel narrowing or knickpoints in the footwall of mapped faults; and iii) other linearly aligned, vertically offset geomorphological sedimentary features such as alluvial fans or landslide deposits. These criteria were linked to active faulting in southern Africa by Jackson and Blenkinsop (1997) and have since been used in recent studies in southern Malawi (Hodge et al., 2018; Wedmore et al., 2020a, b; Williams et al., 2021, 2022).





### 3.1.2 Active Fault Database Attributes

The structure and approach of the LRAFD follows the GEM Global Active Faults Database (GEM-GAFD; Styron and Pagani, 2020). The GEM-GAFD aims to compile attributes relevant to a fault's potential to create an earthquake in a simple structure that contains all information necessary for probabilistic seismic hazard analysis (Christophersen et al., 2015; Styron and Pagani, 2020). Each fault trace is represented by a single GIS feature with a suite of associated attributes. Attributes for the LRAFD were selected from the GEM-GAFD (Styron and Pagani, 2020, their Table 1) with the purpose of describing a fault's

topographic expression and activity confidence (e.g. Styron and Pagani, 2020; Williams et al., 2021, 2022). We do not use all attributes listed in the GEM-GAFD as some are not relevant in this study (e.g. shortening rate) and others (e.g. slip rate; recurrence interval) require analysis (e.g., paleoseismology), which cannot be acquired remotely.

Table 1 lists the attributes of the LRAFD, information about the type of data each attribute represents, and how these attributes are determined. A fault trace represents where a fault is interpreted to have ruptured the ground surface, with each

fault represented by a unique numerical ID. 'Geomorphic Expression' describes the fault trace morphology and the main piece of evidence used to map the fault (Christophersen et al., 2015), and 'Method' distinguishes the dataset used to map each trace. Fault traces more than 5 km apart are mapped as separate features, as these earthquakes are less likely to be able to breach a gap this big (Wesnousky, 2006, 2008). Although some faults may be one continuous structure at depth, we only joined these structures where evidence of linkage is visible at the surface. Consequently, the database includes both discrete faults and sets

of features that may be one fault at depth, but which we have recognized as separate traces based on their surface expression. Exposure and epistemic quality variables are represented by numeric rankings of 1-2. Lower values (1) indicate a high quality of exposure and confidence of faulting. A value of 2 represents a lack of strong fault exposure and reduced certainty a fault exists. There might be strong evidence for an exposed feature on the landscape, but little confidence it is a fault (exposure quality = 1, epistemic quality = 2). Conversely, a fault may have a high confidence of activity but little exposure or representation on

the topography (epistemic quality = 1, exposure quality = 2). Activity confidence is assigned numerically from 1-4: 1 for high confidence and likelihood of recent activity, 4 suggesting the fault shows only weak evidence of activity. Although multiple variables are used to deduce activity confidence, including exposure quality, epistemic quality, and the number of indicators of active faulting, the assigned value remains subjective.

Some mapped faults show limited evidence of recent surface activity, but we include these faults in our database if they

strike between NNE-SSW and ENE-WSW, which means they would be favourably oriented for reactivation given the SE-NW extension direction inferred from focal mechanisms (Delvaux and Barth, 2010) and geodetic models of the motion between the San and Nubian Plates (Wedmore et al., 2021), assuming a moderate fault dip (following Williams et al., 2022). Some major topographical structures may represent inactive faults and therefore, some inactive faults may be included in the LRAFD. As with any active fault database, bias towards inclusiveness reduces the likelihood that potentially active faults are missed (Styron

et al., 2020), but complete mapping of all existing active faults is unlikely, and large earthquake events may occur on unmapped faults.



### 3.1.3 LRAFD Availability and Data Format

The LRAFD is freely available open-source geospatial database containing a collection of active fault traces, and associated attributes in a GIS vector format issued under a Creative Commons (CC-BY-4.0) license. Version 1.0 of the database has been released on the Github and the Zenodo Data Archive: https://doi.org/10.5281/zenodo.6513691. Following the principles outlined in the GEM-GAFD, it is intended that this database will be updated in the future as and when new data comes available to update the attributes associated with the LRAFD and the number of faults in the database. Changes will be made through the Github page (https://github.com/LukeWedmore/luangwa_rift_active_fault_database), and future versions of the database will be released simultaneously on Github and Zenodo when substantial updates to the database are made. The fault database was constructed within ArcGIS, however we have saved the database in several different file formats to aid compatibility with different software and seismic hazard codes. The version of record is the GeoJSON format, which is a plain-text version that can be subject to git version control and is directly compatible with the GEM-GAFD. It is intended that any updates to the LRAFD should be submitted as changes to the GeoJSON file. Other versions of the database are also included on Zenodo and Github in ESRI Shapefile, GeoPackage, KML and GMT formats as well as the conversion script.

## 3.2 Seismogenic Sources in the Luangwa Rift

### 3.2.1 Earthquake Fault Scaling Relationships

Estimates of potential earthquake magnitudes are useful for comparing with historical events and for converting mapped faults into sources for seismic hazard assessment (DISS Working Group, 2021; Williams et al., 2021; Williams et al.). However, since these are estimates are often more subjective and liable to change than the objective and observational data stored in an active fault database, following other similar studies (Faure Walker et al., 2021; Williams et al., 2021), we store these estimates separately from the LRAFD. We use empirical scaling relationships based on fault length (L) to estimate earthquake magnitude ($M_w$), average surface displacement ($D_{av}$), and fault rupture width ($W$), and maximum rupture depth ($MRD$) based on the equations for dip-slip faulting in Leonard (2010):

$$M_w = a \log(L) + b \tag{1}$$

$$\log(D_{av}) = a \log(L) + b \tag{2}$$

$$\log(W) = a \log(L) + b \tag{3}$$

$$MRD = W \sin(\delta) \tag{4}$$

where $\delta$ is fault dip, and $a$ and $b$ are empirically derived constants from Leonard (2010). We propagate uncertainties in $b$ (no uncertainties are provided for $a$ values), and a range of $\delta$ (45°, 53° and 65°) through our calculations. We do not consider multi-fault or segmented ruptures, and hence this modelling assumes that seismicity along faults in the Luangwa Rift resembles the characteristic earthquake model (Schwartz and Coppersmith, 1984).





### 3.2.2 Earthquake Recurrence Intervals and Fault Scarps

We use the 'systems-based' approach outlined in Williams et al. (2021) to derive the recurrence interval and slip rates of the two main border faults in the Luangwa Rift because no palaeoseismic or slip rate studies have been conducted in the region.

Recurrence interval ($R$) is calculated using the equation:

$$R = \frac{d_{av}}{S} \tag{5}$$

where $D_{av}$ is the average displacement (calculated using eq. 2) and $S$ is the slip rate, which is calculated using the following equation:

$$S = \frac{V \, \cos(\theta - \phi)}{\cos \delta} \tag{6}$$

where $\theta$ is the fault slip azimuth, and $V$ and $\phi$ are the horizontal rift extension rate and azimuth. The uncertainties associated with $v$, $\delta$, and $D_{av}$ are propagated through a logic tree (Figure 3), to calculate lower, intermediate, and upper estimates of R. We assume all horizontal extension is accommodated as pure dip-slip motion oriented parallel to the regional extension direction, and thus do not apply uncertainties to $\theta$ and $\phi$. Although eq. 6 raises an apparent inconsistency between faults that are both accommodating dip-slip and oblique to the regional extension direction, this can be explained by local strain

reorientations around faults rooted to deeper-seated weaknesses (Philippon et al., 2015; Williams et al., 2019), and which may be applicable to the Luangwa Rift given that it follows the Mwembeshi shear zone (Sect. 2). We also assume that these border faults accommodate all of the regional extension rate, and thus unlike Williams et al. (2021), we do not weight $V$ or divide it between different faults. This is explored in more detail in the discussion.

### 3.2.3 Fault Segmentation

In the East African Rift, where faults reactivate pre-existing structures, local minima in displacement profiles have been shown be a useful indicator of fault segment boundaries as faults have been suggested to rupture through bends that are often considered geometrical criteria of segmentation (Hodge et al., 2018; Wedmore et al., 2020a). We measured the height of the Quaternary fault scarp for the four faults in the Luangwa Rift with the highest activity confidence (Chipola, Molaza, Kabungo and Chitembo) to identify minima in the along-strike displacement profile that indicate fault segment boundaries. We extracted

across-strike topographic profiles every 30 m, stacked the profiles at 120 m intervals along strike to filter short-wavelength topographic features such as vegetation or human structures that are unrelated to active faulting, and then measured the scarp height across each stacked profile following Wedmore et al. (2020b). We also investigated whether individual topographic profiles showed evidence of multiple earthquakes such as composite scarps and slope breaks (Zhang et al., 1991; Hodge et al., 2020). We assessed histograms of scarp height measurements and looked for bimodal and multimodal distributions that may

suggest the presence of multiple events preserved in the landscape.





## 4 Results

### 4.1 Active Faults and the Luangwa Rift Active Fault Database

The Luangwa Rift Active Fault Database (LRAFD) contains 18 active faults (Table 2; Figure 4). In Figure 8 we use the Chipola Fault to illustrate how we compiled evidence of activity in the LRAFD. The 207 km long Chipola Fault is the longest fault

in the Luangwa Rift and forms the western border fault of the rift basin. The Chipola escarpment and fault scarp are clear even at coarse (1:400000) scale using the DEM, slope, and hillshade maps (Figure 5a-c). The northern end of the fault follows the Karoo-Basement contact (Figure 4c) and a scarp that offsets Quaternary sediments correlates with the mapped faults in the geological maps. The steep scarp (26°) occurs at the base of an abrupt elevation change (Figure 5d) and has displaced Quaternary alluvial fan sediments (Figure 5f). Incised rivers in the faults footwall also suggest recent uplift (Figure 5e). Given

this evidence, the Chipola Fault was assigned 1 for activity confidence, epistemic quality, and exposure quality.

The criteria and indicators listed in Table 1 and described for the Chipola fault above were applied throughout the Luangwa Rift to create the LRAFD. Faults with a similar strength of evidence to that of the Chipola Fault are also mapped with the highest confidence (e.g. the Molaza and Kabungo faults; Figures 6 & 7). Scarps and escarpments are prominent on the DEM and Google Earth (e.g. Figure 5a & 7c), and slope maps highlight steep (>20°) fault scarps that have formed at the base of

many of the escarpments (Figure 5b, 5f, 6, 7b & 7d). Figures 5-7 show slope maps of the Chipola, Mkumpa and Molaza faults, highlighting the steep scarps, with the most prominent (steepest) scarp observed along the Molaza Fault (Figure 6b). Within the Luangwa Rift, these faults show the clearest evidence of offset Quaternary sediments and sedimentary features such as alluvial fans (e.g. Figure 7d-e), indicating recent fault activity and rupture events. We also observed river incision and channel steepening in the footwall of the Mukopa, Chitumbi, Kapampa, Chipola, and Molaza faults (e.g. Figure 6c).

Overall, ten faults had exposure quality scores of 1 indicating they are well exposed, whereas eight faults scored 2 meaning they lacked a strong exposure. Epistemic quality presented 13 traces as high certainty of activity and five as low. Activity confidence was assigned after taking a holistic view of each trace and its likelihood of recent activity. There are six faults with the strongest confidence value (1; Chitumbi, Chipola-South, Chipola, Chitembo, Kabungo and Molaza), and three faults assigned 4 (Chipola-West, Luwi and Mwanya), thought to have a low likelihood of activity.

We found active faults along the length of the 600 km rift, with fault lengths varying between 9 km and 207 km. The faults generally trend NE-SW, with some minor faults trending N-S (Figure 4). Within the rift, the two longest faults, the Chipola Fault (207 km; Figure 5) and the Molaza Fault (142 km; Figure 6) both display evidence of consistent, well-preserved fault scarps, and have the highest activity confidence (1) and exposure quality (1; Table 2). These faults are located at the edge of the rift and represent the western SE-dipping (Chipola; Fault) and eastern NW-dipping (Molaza; Fault) border faults of the rift

(Figure 4). These faults form sub-basins with the Luangwa Rift that differ in their properties. In the northern sub-basin, the Molaza Fault (Figure 6) has only one other active fault within its hangingwall. At the southern end of the Molaza Fault there are two short (9 and 16 km long) faults in a step-over geometry (LRAFD_ID: 14 & 15; Figure 4a), but the rest of the fault displays a relatively simple geometry with no evidence of splays. In the southern sub-basin, where the Chipola Fault is the border fault, there are up to three faults across strike (LRAFD_ID: 8-13; Figure 4a) including two intrarift faults in the centre



of the rift (LRAFD_ID 9 & 10; Figure 4a). In the hanging wall of the northern end of the Chipola Fault, there are two faults (LRAFD_ID: 11 & 12; Figure 4a) in a stepover geometry that are directly across strike from the stepover faults at the southern end of the Molaza Fault. This zone of distributed faulting between the northern and southern sub-basins of the Luangwa Rift, is a common observation in other rift transfer sections in the EAR (Scholz et al., 2020; Kolawole et al., 2021).

### 4.2 Seismic Source Properties

Using the fault scaling laws set out in Leonard (2010), we derived earthquake source parameters including average fault displacement ($D_{av}$), down-dip fault rupture width ($W$), fault maximum rupture depth ($MRD$) and moment magnitude ($M_w$; Table 3). $D_{av}$ varies between 0.3 +0.7/-0.2 m on the Molaza-2 Fault, and 4.2 +9.5/-2.6 m on the Chipola Fault. Predicted fault widths ($W$) range from 8 +3/-1 to 61 +27/-11 km with maximum rupture depths ($MRD$) of 6 +3/-2 to 49 +28/-14 km (Table 3). With these calculations, which assume a reasonable fault dip of 53°, only the Chipola Fault produces a rupture depth that would exceed the crustal thickness of the region (∼45 km; Sun et al., 2021), and only the Chipola and Molaza faults produce a rupture depth that exceeds the maximum depth of seismicity recorded in the region (∼30 km; Craig et al., 2011; Craig and Jackson, 2021). Potential earthquake magnitudes for whole-fault ruptures average $M_w$ 7.0 but vary between $M_w$ 5.8 and 8.1 (Table 3; Figure 8).

### 4.3 Recurrence Intervals

Using eq. 5 and 6 with a logic tree approach (Figure 3; adapted from Williams et al., 2021), we calculated lower, interme-diate, and upper earthquake recurrence intervals ($R$) for the Chipola and Molaza faults (individual logic tress are shown in Supplementary Figure S1 and S2). We applied the rift extension rate ($V$) and azimuth ($\phi$) values of $0.7 \pm 0.2$ mm/yr and 108° from Wedmore et al. (2021). Slip rates for the Chipola and Molaza faults are estimated at 0.4-1.7 mm/yr and 0.4-1.6 mm/yr, respectively. For whole fault ruptures along the Chipola Fault, our intermediate estimate for earthquake recurrence interval is 5,000 years, with a lower estimate of 1,500 years and upper estimate of 36,000 years. For the Molaza Fault the intermediate recurrence interval is estimated at 4,000 years, with a lower bound of 700 years and upper bound of 28,500 years. The large un-certainties associated with these estimates represent the large epistemic uncertainties inherent when propagating uncertainties from empirical scaling relationships to fault recurrence estimates in the systems-based approach (Williams et al., 2021).

### 4.4 Fault Segmentation and Scarp Heights

Example topographic profiles for the 4 faults with the highest confidence of activity (Chipola, Chitembo, Kabungo, and Molaza) are shown in Figure 9, with the corresponding along-strike scarp height profiles in Figure 10. The median scarp height of each fault ranged between $12.9 \pm 0.4$ (Molaza Fault) and $19.2 \pm 0.9$ m (Kabungo Fault – Figure 7 & 10c). The minimum resolvable scarp heights that we were able to measure using the SRTM data was 2-3 m. However, the lower-resolution of SRTM (compared with TanDEM-X - see Wedmore et al., 2020b) meant that we were unable to identify clear fault segment





boundaries. In the statistics quoted below, we have removed outliers from the data, which are values $> 2\sigma$ from the median scarp height.

The SE-dipping Chipola Fault borders the western side of the rift, with a 220 km long fault trace. The median height of the fault scarp is calculated to be $13.0 \pm 0.4$ m. The highest scarps are found towards the southwestern end of the fault, where the maximum scarp height was observed ($28.4 \pm 0.4$ m). The histogram shows that most of the scarp height measurements are

between 5 and 20 m (Figure 10a), and largest peak in the histogram occurring between 14 and 16 m. Smaller peaks are also present at 19 m and 10 m.

The Chitembo Fault trace is 48 km long and dips SE, it is located 20 km East of Chipola's northern tip. The median scarp height is $14.0 \pm 1.0$ m, with a maximum height of $39.6 \pm 0.2$ m (Figure 10b). The histogram of scarp heights peaks between 8-10m, although there is a long tail to the distribution, with other minor peaks observed at 18-20 m and 28-30 m.

The SE dipping 45 km long Kabungo Fault, which is 15 km east of the Chitembo Fault has the highest median scarp height of the faults that we measured $19.2 \pm 0.9$ m. The maximum scarp height is found to be $36.3 \pm 0.08$. The locations where high ($> 30$ m) scarps were observed coincided with where slopes of $>25°$ were observed on the fault scarps, and where a clear change in topography was evident on Google Earth and in the DEM (Figure 7). The histogram of scarp height measurements shows two distinct peaks at 8-10 m and 20-22 m (Figure 10c).

The NW-dipping 142 km long Molaza Fault is the eastern border fault in the northern basin of the Luangwa Rift. The median scarp height is $12.9 \pm 0.4$ m, with a maximum of $30.0 \pm 0.2$ m found at the northern tip, where the slope map shows the most prominent scarp (Figure 6 & 10d). Between 25 and 75 km along strike, there is a consistently preserved scarp averaging $\sim$10 m (e.g. Figure 9d), that coincides with steep slope values of 19°. The histogram of scarp height measurements displays a clear peak at $\sim$10 m with very few exceeding 25 m (Figure 10d).

## 5 Discussion

### 5.1 Characteristics of the LRAFD

We found evidence for Quaternary activity on 18 faults in the Luangwa Rift Zone and developed an active fault database to systematically compile the geomorphic attributes of these faults. This builds on the previous discovery of active faulting along the Chipola South Fault (LRAFD_ID: 5) by Daly et al. (2020). Within the 18 active faults in the Luangwa Rift Active

Fault Database (LRAFD), 6 faults have a very high confidence of recent activity (Table 2). We measured the height of the prominent scarps at the base of the footwall base of two border faults, and two intra-basin faults (Figure 9). Median scarp heights are between 12 and 19 m (Figure 10). Fault scaling relationships suggest that the 207 km long Chipola Fault is capable of hosting earthquakes up to Mw 8.1 with an intermediate recurrence interval estimate of 5055 years and a slip rate of 0.9 mm/yr. The estimated potential earthquake magnitudes exceed previous recorded events in the EARS, but are consistent with

hazard assessments from other active fault and seismic source databases in southern/eastern Africa (Yang and Chen, 2010; Goda et al., 2016; Poggi et al., 2017; Williams et al., 2021). The LRAFD demonstrates that the framework for the future probabilistic seismic hazard analysis in southern Africa outlined by Williams et al. (2021) can be successfully applied to other





regions using freely available, open access data such as SRTM. In addition, the mapped active faults provide an opportunity to analyse the seismotectonics of the Luangwa Rift and compare it to other amagmatic rifts in along the EARS, and this is the

focus of this discussion.

The two main border faults in the Luangwa Rift, the Chipola and Molaza faults, both follow Karoo-Basement contacts, but our analysis shows that these faults have also offset Quaternary sedimentary deposits (Figures 5 & 6). These relationships suggest that these are Karoo age structures that have been reactivated during the current phase of rifting. The Quaternary fault activity adds support to the notion that the southwestern branch of the East African Rift is active, and separates the Nubian

plate from smaller microplates (the San, Rovuma and possibly Angoni microplates) in southern and eastern Africa, as recently demonstrated by geodetic data (Wedmore et al., 2021).

Plate-scale modelling suggests that the extension direction across the San-Nubia plate boundary in the Luangwa Rift is 108 ± 8°(relative to stable Nubia; Wedmore et al., 2021). Focal mechanism inversion shows that the $\sigma3$, minimum compressive stress direction is 123°(Delvaux and Barth, 2010). These extension directions are sub-perpendicular to the fault orientation

found here (mean strike: 045 ± 45°), which follows at the kilometer scale, the orientation of the Mwembeshi shear zone (Figure 2). Both geodetic and seismological data suggest a NW-SE extension direction that is orientated sub-perpendicular to the orientation of the faults in the Luangwa Rift. However, the fault orientations are still consistent with a divergent boundary as we find no geomorphic evidence of horizontal offsets and it is not uncommon for normal faults in the EARS to reactivate at slightly oblique angles to the regional extension direction (Williams et al., 2019). Furthermore, the only available focal

mechanisms from the Luangwa Rift, from a $M_b$ 5.7 earthquake in 1976, shows a normal faulting mechanisms (Nyblade and Langston, 1995). Consequently, we consider that all faults in the LRAFD have pure normal kinematics but note that further work is needed to constrain the stress orientation in this region as the focal mechanism inversion of Delvaux and Barth (2010) is only based on six events, and the geodetic solution of Wedmore et al. (2021) is based on a continental scale GNSS network, with very few stations in the vicinity of the Luangwa Rift. Thus, the LRAFD demonstrates that the Luangwa Rift is an active

rift system that forms the extensional boundary between the Nubia and San plates in southern Africa, with faults that have reactivated Karoo-age structures aligned with the pre-existing lithospheric scale Mwembeshi shear zone.

## 5.2 Fault Activity in the Luangwa Rift and Comparison with other EARS basins

Active and inactive faults are typically distinguished by the age of the most recent earthquakes (Christophersen et al., 2015). However, large magnitude earthquakes do not always result in surface rupture, especially in regions in southern Africa where

the crust can be seismogenic down to 40 km (Jackson and Blenkinsop, 1993; Nyblade and Langston, 1995; Kolawole et al., 2017; Craig and Jackson, 2021; Stevens et al., 2021). Furthermore, we are only aware of two palaeoseismic trenches along the whole of the East African Rift (Kervyn et al., 2006; Zielke and Strecker, 2009; Cohen et al., 2013) that could potentially extend the record of fault ruptures beyond the small number of historically recorded tectonic earthquake surface ruptures (1910 M 7.4 Rukwa, Tanzani - Vittori et al. (1997); 1928 Ms 6.9 Subukia, Kenya - Ambraseys (1991b); 1966 Mw 6.8 Toro, Uganda/DRC

- Loupekine et al. (1966); 2007 Mw 7.0 Mozambique - Fenton and Bommer (2006); Copley et al. (2012); 2009 Karonga sequence, Malawi - Biggs et al. (2010); Macheyeki et al. (2015)). Consequently, it is hard to definitively conclude that a fault



is 'inactive' based on the absence of direct evidence of surface rupture alone. Thus, faults that do not fulfill all active criteria are still included in the database as we applied a broad definition of active faulting to reduce risk of excluding 'inactive' faults that could rupture in a future earthquake, despite displaying limited evidence of activity. In published maps of the region, no
attempt was made to distinguish active and inactive faults in the Luangwa Rift (Banks et al., 1995; Daly et al., 2020). Here we classify 18 faults into varying degrees of activity confidence in a systematic active fault database.

The faults determined to have the highest confidence of activity (Chipola, Molaza, Chitumbi, Kabungo) all have prominent scarps, offset alluvial fans, and steeply incised rivers in the footwall (Figure 5-7). We measured the height of the prominently exposed fault scarps on each of these faults (Figure 9 & 10), with the median scarp height between 13-19 m (Figure 10). The
Chipola Fault has a scarp height of $13.0 \pm 0.4$ m (Figure 9a and 10). It has been suggested that a ~12-14 m high scarp previously detected along the Chipola South Fault formed in the last 10 ka (although no evidence was provided for this time period; Daly et al., 2020), but the authors were unable to distinguish whether this resulted from a single, large magnitude earthquake, or a series of smaller events. Our measurements of scarp height exceed the average single event displacement values from the Leonard (2010) scaling relationships. Thus, our results suggest that these scarps have formed from multiple earthquakes. Along
the Bilila-Mtakataka fault in Malawi, Hodge et al. (2020) demonstrated that a 20 m high fault scarp was generated by at least two earthquakes with single event displacements possibly as high as 10-12 m, which also exceeds empirically derived single event displacement estimates. Evidence from the Hebron Fault in Namibia (Salomon et al., 2021) also suggests that normal faults that rupture thick crust may have higher single event displacement-length ratios than the dataset compiled by Leonard (2010). Alternatively, the scarps may have formed in a single multi-fault event (e.g. Hamling et al., 2017), whose magnitude
(and hence single event displacement) would be higher than indicated in the LRAFD. However, composite scarps observed on the Molaza Fault (Figure 9d), and bi-modal peaks in the histograms of scarp height along the Chipola, Kabugo and Chitembo (Figure 10a-c) support the inference that these scarps have formed in multiple Quaternary earthquakes.

The border faults for the northern (Molaza Fault) and southern (Chipola Fault) basins of the Luangwa Rift have the highest possible values for Activity Confidence, Exposure Quality and Epistemic Quality. Both faults are well exposed along their
entire length (Figures 5 & 6), with steep fault scarps ( 25-30°), and have few gaps where it was not possible to measure the scarp height (Figure 10). In contrast, there are few mapped intra-rift faults. The Luwi and Kapampa faults (LRAFD_ID: 9 & 10) are the most prominent intra-rift faults, and have clear fault scarps, but they have Activity Confidence values of 4 and 3 respectively, and the lowest values for Exposure and Epistemic quality. Furthermore, these intrarift faults are short (Luwi – 20 km; Kapampa – 40 km) compared with the two major border faults (Chipola – 207 km; Molaza – 142 km). The lack of
intrarift faulting is unlikely to be because of the inability to detect smaller scarps with SRTM data as we measured scarps as small as ~3 m high, which are smaller than intrarift faults observed in the southern Malawi Rift (Wedmore et al., 2020b). Thus, deformation in the Luangwa Rift appears to be primarily accommodated across two major border faults at the edge of the rift. This differs from the Malawi Rift where deformation is equally distributed on both border and intra-rift faults (Wedmore et al., 2020b; Shillington et al., 2020), despite both rifts being magma poor and having a similar extension rates (~0.7 mm
yr; Wedmore et al., 2021), and similar crustal (~40-45 km; Sun et al., 2021) and lithospheric thicknesses (~150 km Priestley





et al., 2018). The spatial pattern of deformation in the Luangwa Rift is more similar to the Lake Tanganyika Rift, where up to 90% of extension is accommodated on the border faults (Muirhead et al., 2019).

The localized deformation across border faults justifies our approach of using the geodetically-derived regional extension rate to estimate the slip rate and earthquake recurrence interval of the border faults (Section 3.2.2). However, these should

be treated as upper bounds on the slip rate and lower bounds on the recurrence interval. Numerical models indicate that rifts with localized deformation across border faults form in strong lithosphere, where the strength is dominated by the crust (Huismans and Beaumont, 2011). Low $V_p/V_s$ ratios and high horizontal shear wave velocities suggest the absence of partial melt, magmatic intrusions or significant levels of fluid and instead imply that the crust is strong beneath the Luangwa Rift (Wang et al., 2019; Sun et al., 2021). Furthermore, although faults in the rift follow the orientation of the foliated mylonitic

gneiss and eclogites within Mwembeshi Shear Zone (or a splay of the shear zone Daly et al., 1989), experiments on similar mafic samples from the Malawi Rift suggest these rocks are unlikely to frictionally weak (Hellebrekers et al., 2019). Although the high-grade metamorphic shear zones such as the Mwembeshi Shear Zone are more likely to be viscously weak because of grain-scale heterogeneities, it is notable that Quaternary reactivation of Karoo-age faults in southern Africa has been observed across the Lower Zambezi escarpment in Zimbabwe and in the Lower Shire Rift in Malawi (Mackintosh et al., 2019; Wedmore

et al., 2020a), both regions that are not underlain by lithospheric-scale shear zones. Thus, although the faults in the Luangwa Rift, and other Karoo-age basins have been reactivated during the current Miocene phase of rifting in East Africa, the nature of weaknesses in the lithosphere in this region remains unclear.

## 5.3    5.3 Seismic Source Attributes and Seismic Hazard Implications

The largest historical event in the LRZ was a $M_w$ 6.7 event on 1st May 1919 (Figure 8; International Seismological Centre,

2021,  NEIC Earthquake Catalogue). The ISC-GEM catalogue indicates that another $M_w$ 6.3 event occurred in the same region in 1940 (Figure 6). The epicentres of both the 1919 and 1940 events are located close to a ~50 km long section of the Molaza Fault that is exceptionally prominent, with a well-preserved, linear fault scarp that is continuous across small stream channels (Figure 6). Empirical fault scaling laws imply that the $M_w$ 6.7 would cause a 30 km long rupture with an average displacement of 0.9 m, and the $M_w$ 6.3 event would cause a 17 km long rupture Leonard (2010). Thus, we suggest that the ~50 km long

exceptionally well-preserved fault scarp along the Molaza Fault was formed, in part, by the two 20th century earthquakes in the Luangwa Rift. However, earthquake location accuracy in Africa at the time of these events is low, shown by the disparity between NEIC and ISC-GEM locations (Figure 6). The 1919 earthquake recorded by ISC-GEM occurs on the same day as a $M_s$ 6.2 event recorded on 1st May 1919, which macroseismic damage reports initially suggest was located  250 km to the north (Ambraseys and Adams, 1991). It is unclear if these events are linked. Nevertheless, previous seismic hazard assessment in the

region by definition considers the maximum possible earthquake magnitude to be 0.5 greater than the largest recorded historical earthquake (Poggi et al., 2017). However, this seismic hazard assessment states that the maximum magnitude earthquake in this region is M 6.9 (Poggi et al., 2017). Our new finding of a $M_w$ 6.7 event on the Molaza Fault should therefore prompt a revision of the seismic hazard in the region.





Despite evidence for the activity on the Molaza Fault in the 20th century, there remains large portions of the 140 km long
fault that have not ruptured recently. We estimate that the Molaza Fault is a seismic source capable of hosting a Mw 7.8
earthquake with a displacement of 3.1 m, which is an order of magnitude greater than earthquakes recorded in the Luangwa
region and more than the any event in the whole of southern/eastern Africa. The largest regional event was the 13th December
1910 Ms 7.4 Rukwa earthquake in Tanzania (Ambraseys, 1991a) and there have only been five other M≥7.0 events along
the East African Rift: the 1919 Mw 7.2 Matai, Tanzania, earthquake; the 1928 $M_w$ 7.0 Baringo, Kenya, earthquake; the
two Juba, Sudan, earthquakes in 1990 ($M_w$ 7.1 and 7.2; Girdler and McConnell, 1994) and the 2006 $M_w$ 7.0 Mozambique
earthquake (Copley et al., 2012). The LRAFD contains eight faults that exceed 50 km in length, with two faults greater than
100 km. Seismic source attributes calculated from the LRAFD indicate that there are 12 faults that have the potential to rupture
in $M_w$ ≥7.0 earthquakes (Figure 8), with the 207 km long Chipola fault capable of hosting a $M_w$ 8.1 earthquake. Global
compilations of continental normal faulting earthquakes suggest that they rarely exceed rupture lengths of 50 km and low $M_w$
7 (Neely and Stein, 2021), and there is only one event with a surface rupture length > 100km (Valentini et al., 2020). Thus,
although $M_w$ 7+ events are likely rare, they should be considered possible in the Luangwa Rift due to the long faults that are
hosted in 45 km thick crust (Sun et al., 2021), which has recorded seismicity to ∼30 km depth (Craig et al., 2011; Craig and
Jackson, 2021). Nonetheless, these large magnitude events likely occur infrequently as >100 km long normal faults in southern
Africa are often segmented (Mortimer et al., 2016; Hodge et al., 2018; Wedmore et al., 2020a, b) and the regional b-value is
∼1 (Poggi et al., 2017) implying that smaller segmented ruptures are more likely than earthquakes that rupture an entire fault.

We attempted to identify fault segments for the four best exposed faults in the Luangwa Rift by systematically measuring
along-strike fault scarp heights (Figure 10). This approach has been successful in other East African rift basins using 12.5 m
resolution TanDEM-X data (Hodge et al., 2018; Wedmore et al., 2020a, b). Using the 30 m resolution SRTM data, we were
unable to identify clear segment boundaries in either the scarp height measurements or in notable changes in fault geometry
(e.g. 90°bends), making it challenging to assess the limits on rupture length on individual faults. The low slip rates that we
calculated for these faults (0.4-1.7 mm/yr) indicate that if large (M>7.0) events do occur, they are rare, with intermediate
earthquake recurrence intervals of 5.1 ka on the Chipola Fault and 3.4 ka on the Molaza Fault. Thus, Although we do not
directly observe evidence for fault segmentation in the Luangwa Rift, the data provided here can be used to incorporate small
ruptures along these faults into seismic hazard assessment by combining the provided slip rate, fault area, and magnitude
estimates with a regional b-value (Poggi et al., 2017) and the methodology developed by Youngs and Coppersmith (1985)
to develop continuous recurrence models for these sources (see Williams et al.). However, it is also possible that multi-fault
rupture may occur, which would increase the potential magnitude of future events. For example, multi-fault rupture of the 142
km long Molaza (LRAFD_ID: 16), 9 km long Molaza 2 (LRAFD_ID: 15) and 16 km long Molaza 3 (LRAFD_ID: 14) faults
would increase the potential magnitude earthquake $M_w$ 7.8 to 8.0 compared to if the Molaza fault ruptured on its own.

The low frequency of large earthquakes in the Luangwa Rift and few recent destructive earthquakes means that awareness of
seismic hazard and mitigation strategies in Zambia may be low. The Luangwa Valley and National Park are tourist destinations
with a significant number of communities, tourists, wildlife, and economy exposed to seismic hazard. Kasama (200 km west)
and Chipata (150 km east) are the largest nearby cities, both with populations of ∼90,000 (Zambia Statistics Agency, 2022).



There are a combined 2.4 million people living in the Muchinga and Eastern regions of Zambia, with a population growth
rate of 4.3 % and 2.8% respectively (Zambia Statistics Agency, 2022). This means nearby populations that may be affected by
seismic hazard will increase with time. Furthermore, the region has high levels of vulnerability. Recent moderate magnitude
earthquakes in Malawi led to high levels of damage and large economic losses (World Bank, 2019) and research in Malawi
indicates that building vulnerability in this region is higher than currently predicted by global models (e.g. the USGS WHE-
PAGER model; Novelli et al., 2021; Giordano et al., 2021). We suggest that active fault mapping, such as has been carried out
here in the LRAFD and in other active fault databases in southern Africa (Williams et al., 2021, 2022) provides a framework
for accurate probabilistic seismic hazard assessment, and thus for increasing resilience to seismic hazard throughout southern
and eastern Africa.

## 6 Conclusions

Using SRTM data, we have systematically mapped and compiled the attributes of 18 known active faults in the Luangwa Rift,
Zambia to produce the Luangwa Rift Active Fault Database (LRAFD). The LRAFD is freely available open-source dataset that
is aimed for use in future probabilistic seismic hazard assessment, as well as providing a resource of further scientific study of
the Luangwa Rift. Empirical scaling relationships between fault length and earthquake magnitude suggest that the faults in the
Luangwa Rift can host earthquakes greater than $M_w$ 7, up to $M_w$ 8.1, although we consider that these scenarios are unlikely or
extremely rare.

We find evidence that all 18 faults mapped have been active during the Quaternary, with the four most prominent faults
displaying well preserved linear fault scarps up to ∼20-30 m high. Systematic measurements of the height of the scarps on
these four faults suggest that they were formed by multiple earthquakes, but using 30 m resolution SRTM data, we were unable
to use along-strike scarp height profiles to identify fault segment boundaries. Within the Luangwa Rift, the two border faults
(Chipola and Molaza), which have opposing polarity and have reactivated structures that were previously active during a Karoo
phase of rifting, appear to accommodate most of the surface deformation. This suggests that the 45 km thick crust is strong
and does not contain any weaker mid- or lower-crustal layers, which is confirmed by other geophysical proxies. Although the
orientation of the faults in the rift follows that of the underlying Mwembeshi shear zone, it remains unclear why this shear zone
is weaker than the surrounding rocks. Nevertheless, we conclusively demonstrate that faults in the Luangwa Rift are active,
and provide a pathway for the inclusion of active faults in the region into future probabilistic seismic hazard assessment.

*Data availability.* All data generated in this manuscript is freely available and archived in online repositories. Version 1 of The Luangwa
Rift Active Fault Database (LRAFD) is available at: https://doi.org/10.5281/zenodo.6513691. The LRAFD is also available on github (https:
//github.com/LukeWedmore/luangwa_rift_active_fault_database) and we encourage authors to suggest future changes and additions to the
database through the github repository. The seismogenic source parameters are available at: https://doi.org/10.5281/zenodo.6513778. The
measurements of scarp height are available at: https://doi.org/10.5281/zenodo.6513545.



475  *Author contributions.* This project was devised by Luke Wedmore and Juliet Biggs as part of Tess Turner's MSci project. Tess Turner, Luke Wedmore and Jack Williams carried out the fault mapping, Tess calculated the seismogenic source properties and measured the scarp heights using codes developed by Luke Wedmore. Luke Wedmore developed and archived the active fault database in collaboration with all co-authors. Luke Wedmore wrote the manuscript with assistance from all co-authors.

*Competing interests.* The authors are not aware of any competing interests.

480  *Acknowledgements.* This manuscript originated as a University of Bristol MSci project by Tess Turner. The overall project was funded by several grants under EPSRC Global Challenges Research Fund (GCRF): PREPARE (EP/P028233/1); SAFER PREPARED (part of the 'Innovative data services for aquaculture, seismic resilience and drought adaptation in East Africa' grant; EP/T015462/1); and a GCRF EPSRC Institutional Sponsorship Award. This work greatly benefited from discussions with Åke Fagereng throughout the course of the PREPARE project.



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



**Table 1.** The Luangwa Rift Active Fault Database attributes. The attributes are adapted from the Global Earthquake Model Global Active Faults Database (Styron and Pagani, 2020).

| Attribute | Type | Description | Notes |
|---|---|---|---|
| LRAFD_ID | integer | Unique Fault IDentification number assigned to each fault trace | |
| Fault_Name | string | Name of fault | Assigned using local geographic features or towns |
| Dip_Direction | string | Compass quadrant of fault dip direction | |
| Slip_type | string | Kinematic type of fault | e.g. normal, reverse, sinistral-strike slip, dextral-strike slip |
| Fault_Length | decimal | Straight line distance between fault tips | measured in km |
| GeomorphicExpression | string | Geomorphic feature/features used to identify the fault trace and its extent | e.g., escarpment, scarp, offset sedimentary features |
| Method | string | DEM or geologic dataset used identify and map the fault trace | e.g., digital elevation model hillshade, slope map |
| Confidence | integer | Confidence of Quaternary activity | Ranges from 1-4, 1 if high certainty and 4 if low certainty |
| ExposureQuality | integer | Fault exposure quality | 1 if high, 2 if low |
| EpistemicQuality | integer | Certainty that fault exists here | 1 if high, 2 if low |
| Accuracy | integer | Coarsest scale at which fault trace can be mapped, expressed as the denominator of the map scale | Reflects the prominence of the fault's geomorphologic expression |
| GeologicalMapExpression | string | Extent of correlation between fault traces and geological maps | Whether faults have been previously mapped and/or follow geological contacts |
| Notes | string | Any additional or relevant important regarding the fault | |
| References | string | Relevant literature/geological maps where faults have been previously mentioned/described | |





**Table 2.** Abridged version of the Luangwa Rift Active Fault Database (LRAFD). The full version of the database can be accessed at: https://doi.org/10.5281/zenodo.6513691 (Wedmore et al., 2022). See Table 1 for a description of the attributes.

| LRAFD_ID | Fault_Name | Dip_Direction | Fault_Length | Confidence | ExposureQuality | EpistemicQuality |
|---|---|---|---|---|---|---|
| 1 | Mulungwe | SE | 48 | 2 | 1 | 1 |
| 2 | Mukopa | SE | 45 | 2 | 1 | 1 |
| 3 | Chitumbi | SE | 85 | 1 | 1 | 1 |
| 4 | Kaloko | SE | 25 | 3 | 2 | 2 |
| 5 | Chipola-S | SE | 54 | 1 | 1 | 1 |
| 6 | Chipola-W | SE | 32 | 4 | 1 | 2 |
| 7 | Chipola | SE | 207 | 1 | 1 | 1 |
| 8 | Mkumpa | W | 52 | 3 | 2 | 1 |
| 9 | Luwi | E | 20 | 4 | 2 | 2 |
| 10 | Kapampa | SE | 40 | 3 | 2 | 2 |
| 11 | Chitembo | SE | 48 | 1 | 1 | 1 |
| 12 | Kabungo | SE | 45 | 1 | 1 | 1 |
| 13 | Mwanya | NW | 65 | 4 | 2 | 2 |
| 14 | Molaza-3 | NW | 16 | 3 | 2 | 1 |
| 15 | Molaza-2 | NW | 9 | 3 | 2 | 1 |
| 16 | Molaza | NW | 142 | 1 | 1 | 1 |
| 17 | Kuta | NW | 87 | 2 | 1 | 1 |
| 18 | Musamba | E | 65 | 3 | 2 | 1 |



**Table 3.** Seismogenic source properties for faults in the LRAFD, calculated using the fault scaling laws set out in Leonard (2010). https://doi.org/10.5281/zenodo.6513778 (Turner et al., 2022b). The length ($L$) of each fault is measured in a straight line between the two tips; estimated earthquake magnitude ($M_w$) from applying Eq. 1; displacement ($D_{av}$, m) using Eq. 2; width ($W$) calculated using Eq. 3; and maximum rupture depth ($MRD$) using eq. 4 and the intermediate dip value of $53°$. The uncertainties for each parameter are calculated by propagating the empirical uncertainties from Leonard (2010)

| LRAFD_ID | Fault Name | $L$ (km) | $M_w$ | $D_{av}$ (m) | $W$ (km) | $MRD$ (km) |
|---|---|---|---|---|---|---|
| 1 | Mulungwe | 48 | 7.0 | $1.3^{+2.8}_{-0.8}$ | $23^{+10}_{-4}$ | $18^{+10}_{-5}$ |
| 2 | Mukopa | 45 | 7.0 | $1.2^{+2.7}_{-0.7}$ | $22^{+10}_{-4}$ | $18^{+10}_{-5}$ |
| 3 | Chitumbi | 85 | 7.5 | $2.0^{+4.5}_{-1.2}$ | $34^{+15}_{-6}$ | $27^{+15}_{-8}$ |
| 4 | Kaloko | 25 | 6.6 | $0.7^{+1.6}_{-0.4}$ | $15^{+7}_{-3}$ | $12^{+7}_{-3}$ |
| 5 | Chipola-S | 54 | 7.1 | $1.4^{+3.1}_{-0.8}$ | $25^{+11}_{-5}$ | $20^{+11}_{-6}$ |
| 6 | Chipola-W | 32 | 6.8 | $0.9^{+2.0}_{-0.5}$ | $18^{+8}_{-3}$ | $14^{+8}_{-4}$ |
| 7 | Chipola | 207 | 8.1 | $4.2^{+9.5}_{-2.6}$ | $61^{+27}_{-11}$ | $49^{+28}_{-14}$ |
| 8 | Mkumpa | 52 | 7.1 | $1.3^{+3.0}_{-0.8}$ | $24^{+11}_{-5}$ | $19^{+11}_{-5}$ |
| 9 | Luwi | 20 | 6.4 | $0.6^{+1.4}_{-0.4}$ | $13^{+6}_{-2}$ | $10^{+6}_{-3}$ |
| 10 | Kapampa | 40 | 6.9 | $1.1^{+2.4}_{-0.7}$ | $20^{+9}_{-4}$ | $16^{+9}_{-5}$ |
| 11 | Chitembo | 48 | 7.0 | $1.3^{+2.8}_{-0.8}$ | $23^{+10}_{-4}$ | $18^{+10}_{-5}$ |
| 12 | Kabungo | 45 | 7.0 | $1.2^{+2.7}_{-0.7}$ | $22^{+10}_{-4}$ | $18^{+10}_{-5}$ |
| 13 | Mwanya | 65 | 7.3 | $1.6^{+3.6}_{-1.0}$ | $28^{+13}_{-5}$ | $23^{+13}_{-6}$ |
| 14 | Molaza-3 | 16 | 6.3 | $0.5^{+1.1}_{-0.3}$ | $11^{+5}_{-2}$ | $9^{+5}_{-2}$ |
| 15 | Molaza-2 | 9 | 5.8 | $0.3^{+0.7}_{-0.2}$ | $8^{+3}_{-1}$ | $6^{+3}_{-2}$ |
| 16 | Molaza | 142 | 7.8 | $3.1^{+6.9}_{-1.9}$ | $47^{+21}_{-9}$ | $38^{+22}_{-11}$ |
| 17 | Kuta | 87 | 7.5 | $2.1^{+4.6}_{-1.3}$ | $34^{+15}_{-6}$ | $27^{+16}_{-8}$ |
| 18 | Musamba | 65 | 7.3 | $1.6^{+3.6}_{-1.0}$ | $28^{+13}_{-5}$ | $23^{+13}_{-6}$ |





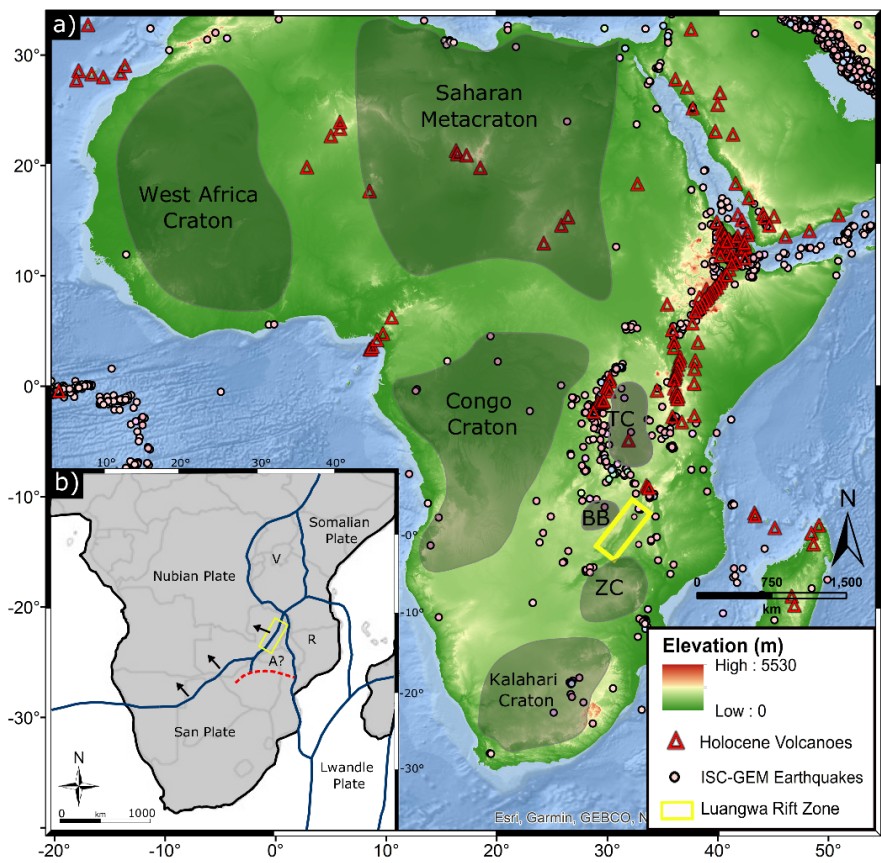

**Figure 1.** The seismicity, volcanism, and plate boundaries of the EARS. (a) Seismotectonic map that highlights the topography of the African continent, the locations of Archean cratons (adapted from Gubanov and Mooney, 2009), and the prevalence of earthquakes in the Proterozoic belts surrounding the cratons. TC – Tanzania Craton, ZC – Zimbabwe Craton, BB- Bangweulu Block. Holocene active volcanoes are represented by red triangles and earthquake epicentres (circles) from the ISC-GEM catalogue. The Luangwa Rift is highlighted in yellow. (b) Plate configuration of southern Africa including the Victoria (V), Rovuma (R), Lwandle, and San microplates (Calais et al., 2006; Stamps et al., 2008; Wedmore et al., 2021) and the model proposed by (Daly et al., 2020) Daly et al. (2020) where the Angoni microplate (A) is separate from San. The black arrows are vectors of the Nubian plate with respect to San, which are fault and earthquake defined (Daly et al., 2020).



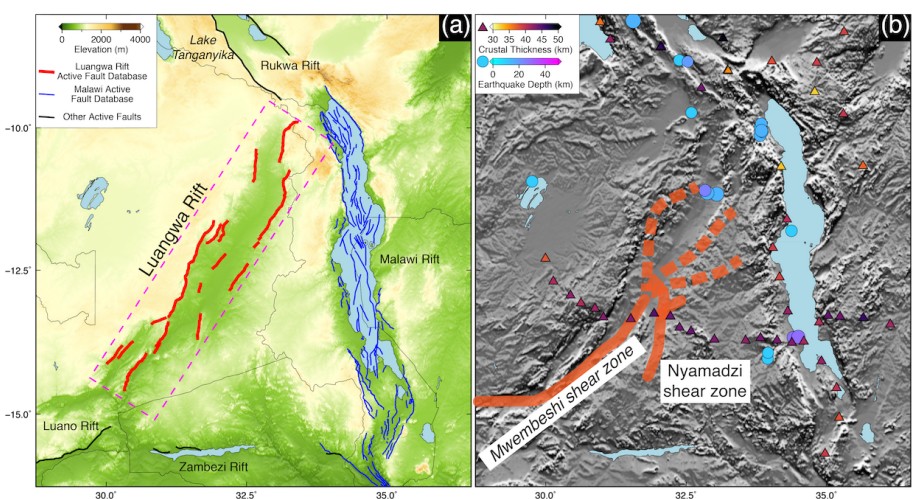

**Figure 2.** The Luangwa Rift and the intersection between southwestern and western branches of the EARS showing major faults. (a) The faults in the LRAFD are shown with red lines, the Malawi Active Fault Database (MAFD) active fault traces in blue lines (Williams et al., 2022), and other active border faults are outlined in black (Chorowicz, 2005). (b) Seismicity ($M_w$>5.2 from the ISC-GEM catalogue), crustal thickness (from receiver function measurements; Sun et al., 2021), and the location of major lithospheric-scale shear zones beneath the Luangwa Rift (adapted from Daly et al., 1989; Johnson et al., 2006; Alessio et al., 2019).





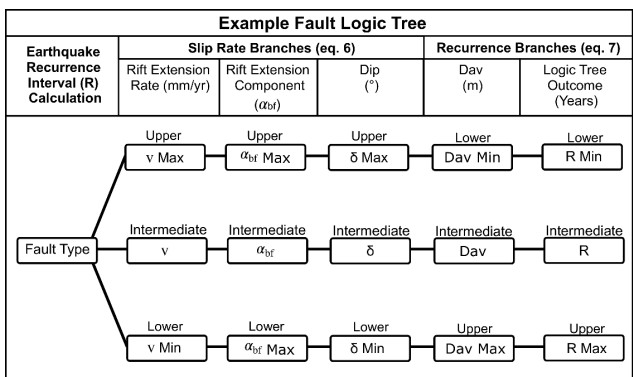

**Figure 3.** Logic tree used to calculate recurrence intervals for the Chipola and Molaza faults (adapted from Williams et al., 2021). Lower, intermediate and upper estimates of fault dip, displacement and horizontal rift extension rate are used, but we do not apply weightings to the component of rift extension rate taken up by the rift border fault ($\alpha_{bf} = 1$) as there are insufficient constraints on the strain accommodation of border and intrarift faults in the Luangwa Rift.





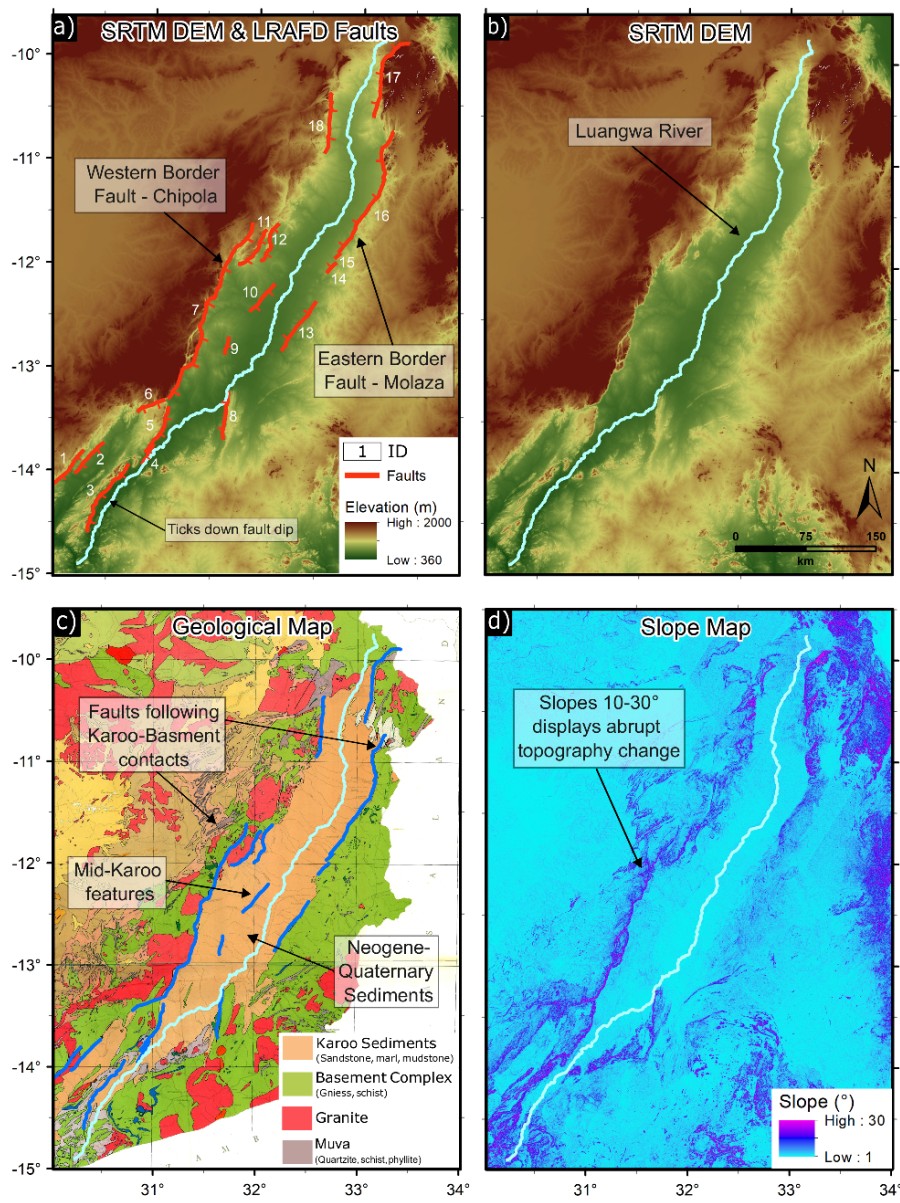

**Figure 4.** Topography and geology of the Luangwa Rift. Each figure represents a different technique used to map fault traces, denoted in the methods attribute of the LRAFD. (a) SRTM DEM of the rift with the 18 fault traces shown in red, the dip direction of each fault is indicated with the dash direction. (b) SRTM DEM without the fault traces. (c) Georeferenced geological map of the Luangwa Rift (adapted from Priday and Camps, 1960) overlain with fault traces in blue. Karoo sediments are overlain by Neogene-Quaternary sediments, not represented on the map (Utting, 1988; Banks et al., 1995; Bishop et al., 2016). Faults predominantly follow contacts between Karoo sediments and Basement complex, representing zones of weakness. (d) Slope map of the rift where increased slope values correspond to the steep fault scarps which were used to map traces.



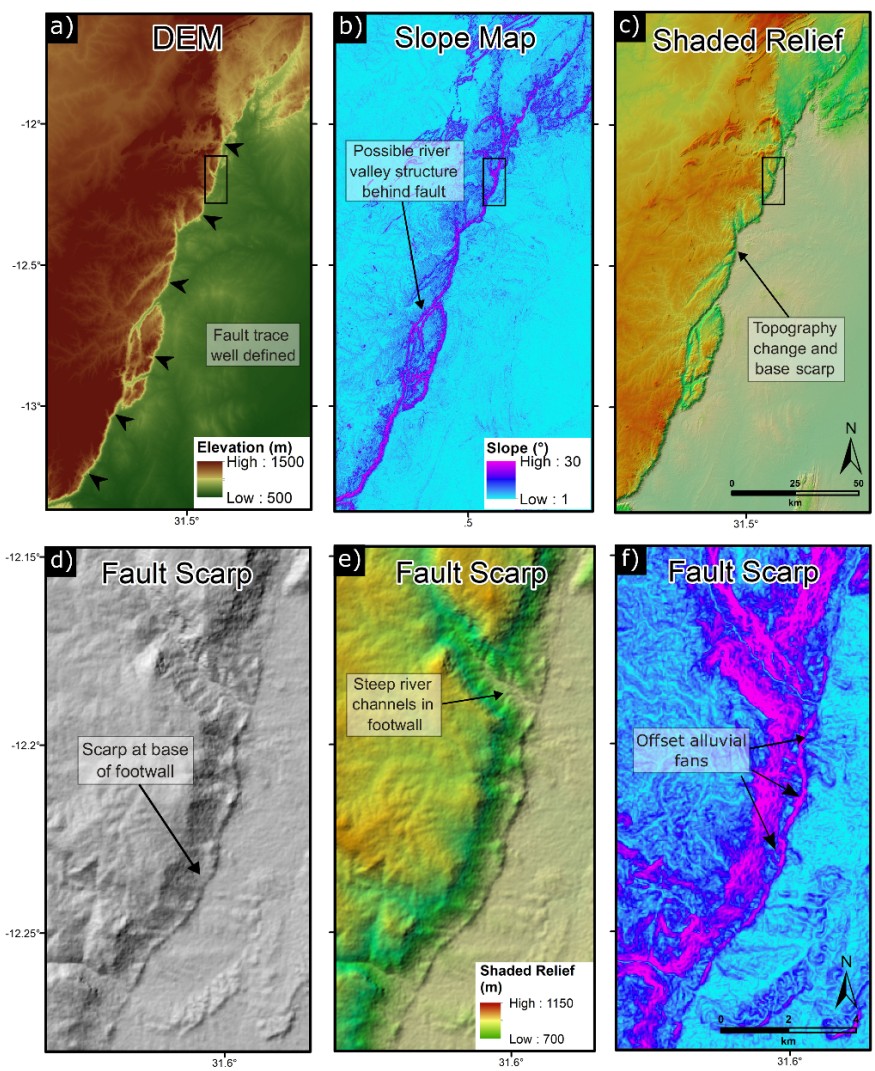

**Figure 5.** The Chipola Fault. (a-c) Overview of the Chipola fault shown in the DEM, slope and shaded relief maps adapted from SRTM data (Farr et al., 2007). The fault is notable for the steep ( 25-30°) fault scarp at the base of the ~1000 m high escarpment. (d-f) Hillshade, shaded relief and slope maps showing a zoomed in section of the Chipola fault indicating the Quaternary fault scarp that has formed at the base of the escarpment and that has offset alluvial fan deposits.



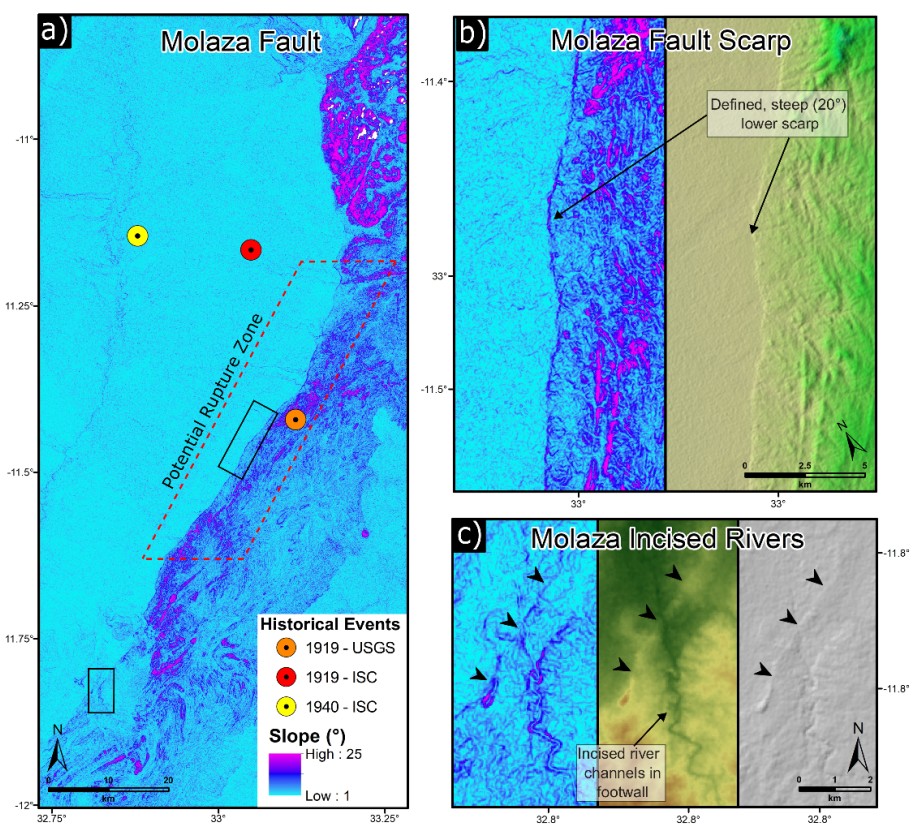

**Figure 6.** Evidence of Quaternary activity along the Molaza Fault. (a) Slope map of the Molaza Fault overlaid with different estimates of the epicentre of two major 20th century earthquakes that occurred within the rift on 1st May 1919 (M 6.7) and 1940 ($M_w$6.3). Source depth for both events is essentially unknown (quoted as 15 km depth on both catalogues). Both epicentres coincide with a portion of the fault where the Quaternary scarp is consistently preserved (part b). Further evidence of activity is provided by steeply incised rivers in the footwall of the fault (part c).



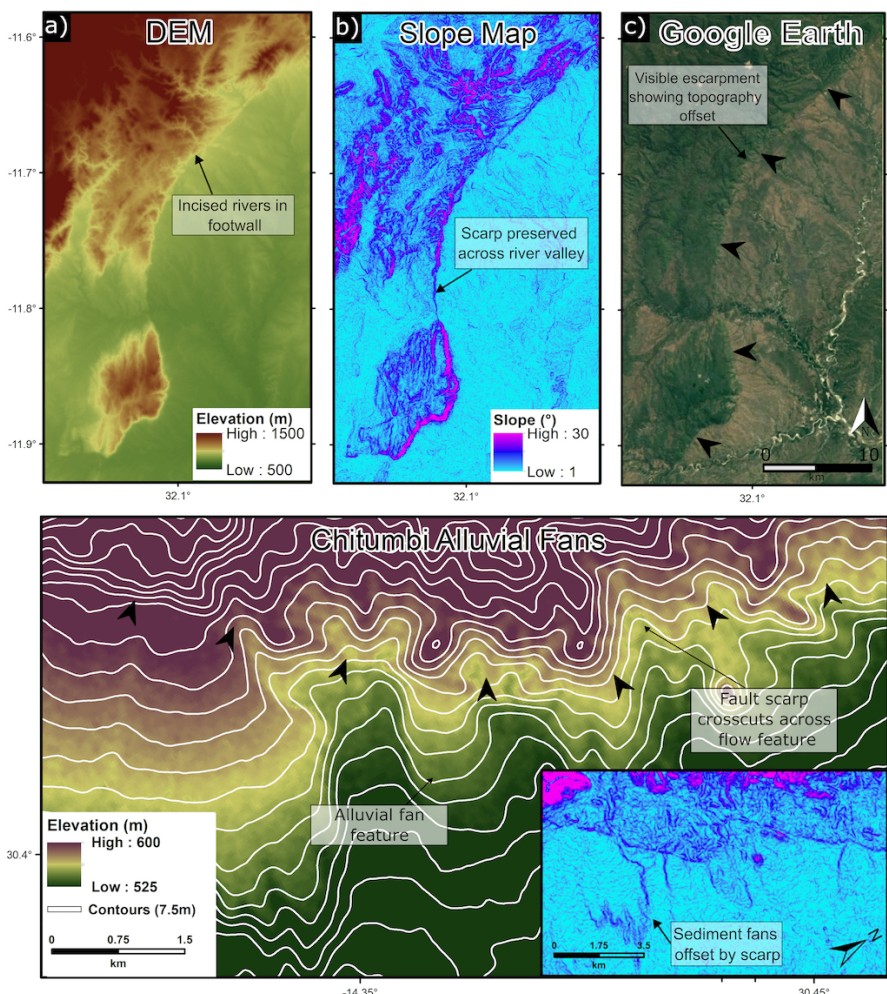

**Figure 7.** Example of active fault indicators on the Kabungo Fault. (a-c) Slope, DEM, and google earth maps (©Google Earth) show the pronounced Kabungo fault escarpment, with a steep (30°) scarp. The scarp is preserved across the river channel, indicating a recent offset such that the channel has not had time to fully respond after uplift. (d) Contour map of alluvial fans offset by the Chitumbi Fault (slope map of the same area is inset).



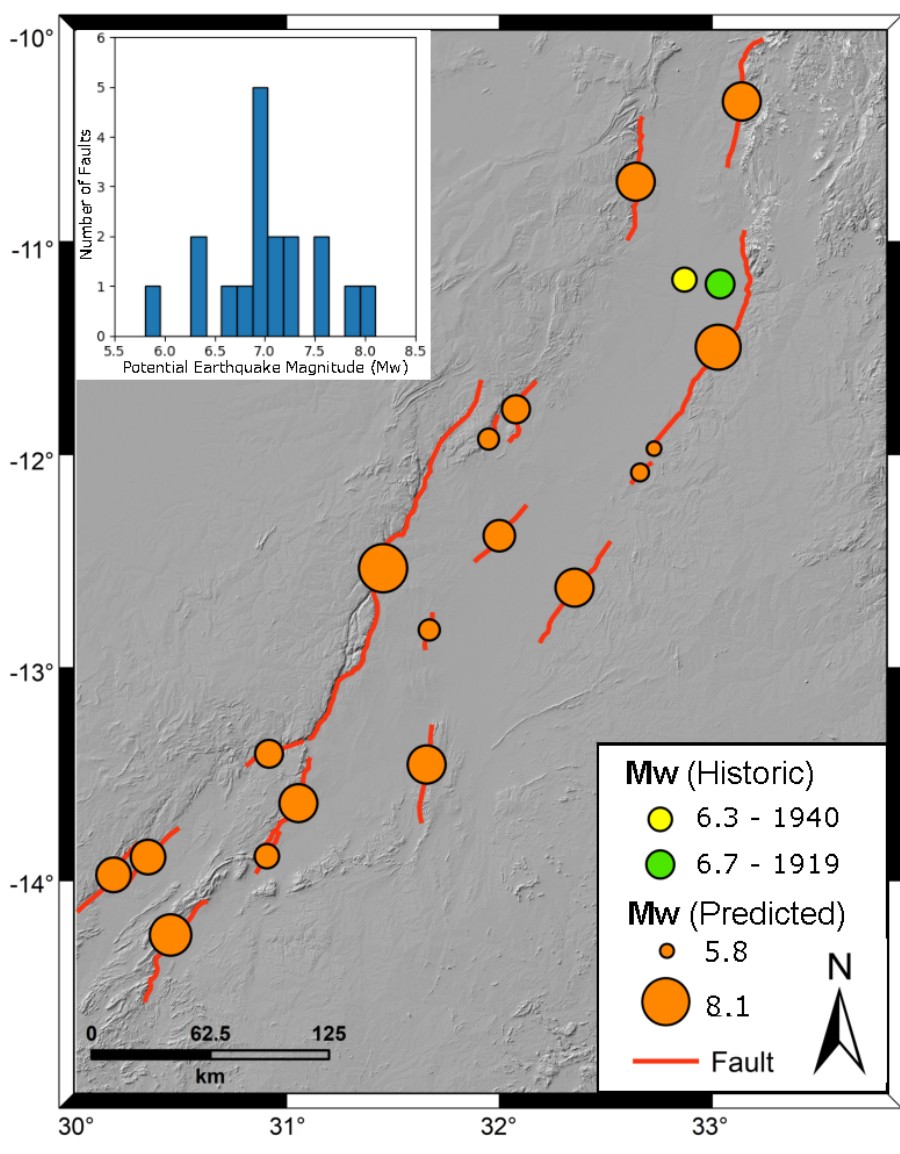

**Figure 8.** Hillshade map of the Luangwa Rift showing estimated maximum earthquake magnitudes for each fault (orange circles) alongside previous recorded seismicity. Inset shows a histogram of maximum potential magnitudes (estimated using empirical relationships between fault length and earthquake magnitude), which range from $M_w$ 5.8-8.1.



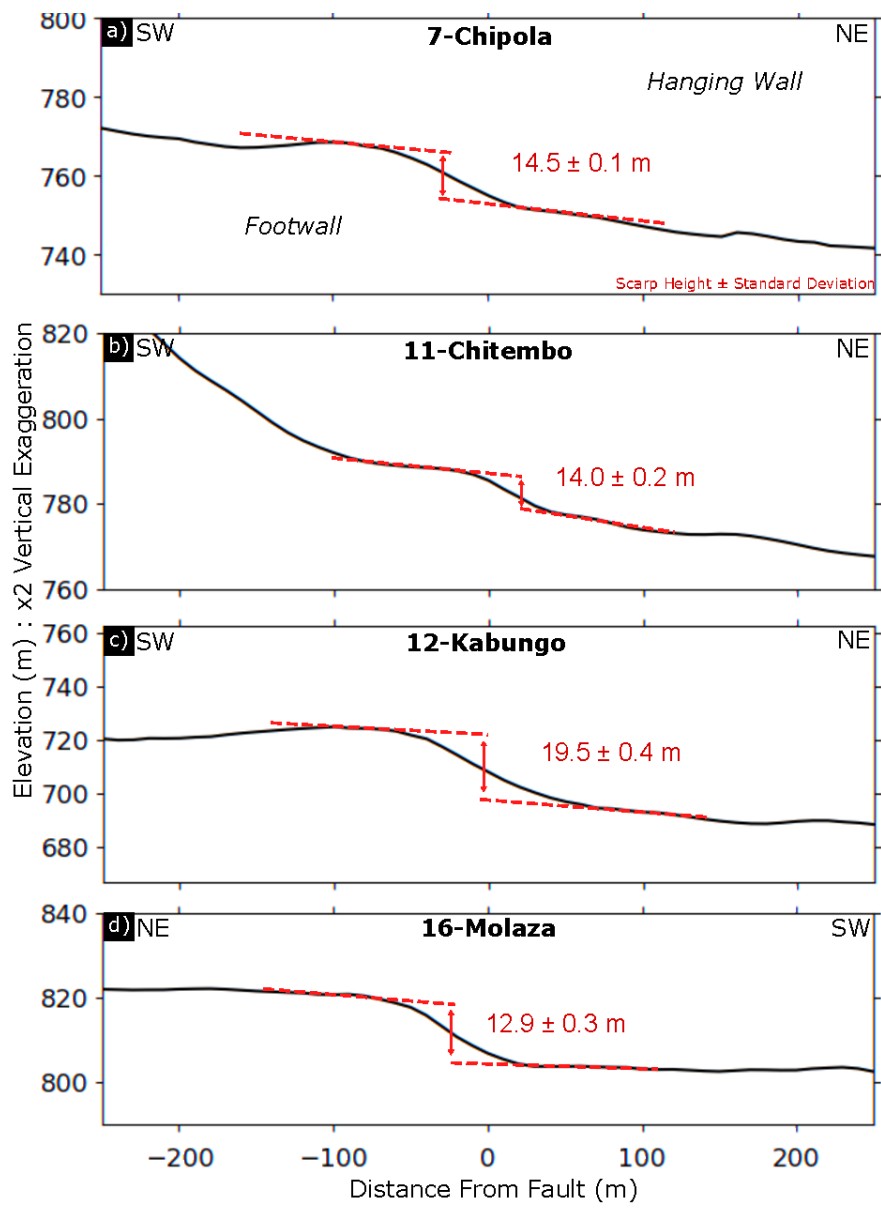

**Figure 9.** Example topographic profiles measured across faults in the Luangwa Rift showing the Chipola (a), Chitembo (b), Kabungo (c), and Molaza (d) faults.





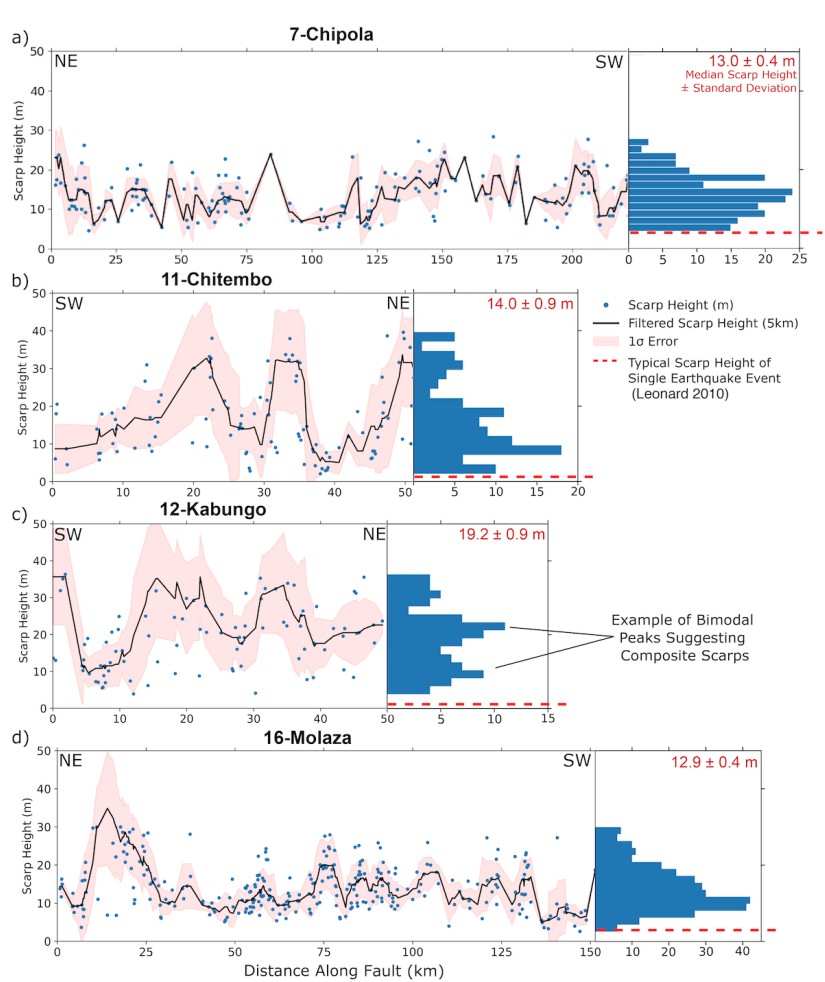

**Figure 10.** Along-strike fault scarp height profiles for 4 faults in the Luangwa Rift. Horizontal axis is not to scale but represent relative variations in fault length. Circles represent individual scarp height measurements, black lines show the 3 km filtered moving median offset, and pink shading shows the $1\sigma$ error. Histograms represent the frequency of scarp heights in the profile. Median scarp heights are shown in red with the uncertainty the standard error. The red dashed lines represent the average single event displacement ($D_{av}$) values for each fault derived from empirical relationships (Leonard, 2010). Scarp heights exceeding this, along with bimodal histogram peaks, suggest the potential for multiple earthquake events and composite scarps. This data has been archived and is available at the following location: https://doi.org/10.5281/zenodo.6513545 (Turner et al., 2022a).