# Peer review of "The Luangwa Rift Active Fault Database and fault reactivation along the southwestern branch of the East African Rift"

_EGUsphere, 2022_

## Author Comment (AC1)

Dear Editor,

We thank the reviewers for a constructive set of reviews that will improve the manuscript. We have responded in detail to them in the attached document, and included text that we would include in the revised manuscript.

Thanks,

Luke Wedmore (on behalf of co-authors).

Our responses to the reviewer 1's comments are shown in blue text. **Specific changes that we will make to the text are in bold font.** The line numbers quoted refer to the revised version of the manuscript that we will resubmit if invited.

RC1

1. An active fault has been widely defined via offset late Quaternary landforms or sediments. In this study, no field observation on these faults can show any evidence for the latest activities. For large earthquakes, we can often find coseismic surface ruptures. There is a historic earthquake in the 20[th] Some records in the history may help you.

We acknowledge that 'late Quaternary' was mentioned in the abstract and this was overly specific – **We will remove any reference to the term 'late Quaternary'** and stick with the broader Quaternary definition. In our methods section (3.1.1 Mapping Faults), we make it clear that the age of the sediments that have been offset by these faults is poorly determined, but that a Quaternary age is consistent with the age of dated sediments within the Luangwa Rift (Barham, 2011; Banks, 1995), and with the age of other rifts in southern Africa (e.g. Malawi, Scholz et al 2007).

In the discussion section (section 5.3) we discuss 20[th] century earthquakes that have been located within the Luangwa Rift, in particular in close proximity to the Molaza Fault. From the locations given by the USGS and ISC-GEM catalogue, we believe it is highly likely that these earthquakes caused a surface rupture on the Molaza Fault which is stated in the text:

"The epicentres of both the 1919 and 1940 events are located close to a ~50 km long section of the Molaza Fault that is exceptionally prominent, with a well-preserved, steep (20-25°), linear fault scarp that is continuous across small stream channels (Figure 6)… Thus, we suggest that the ~50 km long exceptionally well-preserved fault scarp along the Molaza Fault was formed, in part, by the two 20th century earthquakes in the Luangwa Rift." [Lines 417-422; Section 5.3]

However, we also caveat this because location accuracy of earthquakes in southern Africa during the early-mid 20[th] century is very poor. Furthermore, as we have been unable to conduct fieldwork in the Luangwa Rift, we cannot verify this with surface observations – but this should be a priority for future research. Therefore, although we strongly suspect we have discovered a coseismic surface rupture from two 20[th] century earthquakes, and say so in the text, we feel it is important for balance to identify the uncertainties related to this.

**We will add the following text to emphasise that ground-truthing the rupture should be a priority:**

*"field investigation of the Molaza Fault should be a priority to establish whether this represents a rare example of a 20th century earthquake surface rupture in Africa."* **[lines 425-427; section 5.3].**

2. You suggest the fault scarps on the alluvial fan. Do you have any evidence for the fan age and the same geomorphic units on the both sides of the fault?

As mentioned in response to point #1, unfortunately, we do not have evidence for the specific age of the faulted alluvial fan, other than that it likely formed during the Quaternary. However, we were careful to ensure that the same geomorphic units were on both sides of the fault. We did this by plotting the slope map (e.g. Figure 5), to ensure similar slope values either side of the fault, and by plotting topography contours (Figure 7).

**We will amend the text to clarify this:**

*"The criteria were assessed by looking at the DEM, geologic maps and by calculating derived products including slope and hillshade maps from the SRTM data…. By looking for similar topographic gradients either side of the fault, slope maps were particularly useful to ensure that on offset sedimentary landforms, the same geomorphic units were present on either side of the fault."* **[lines 117-122; section 3.1.1]**

3. The magnitudes are estimated via the empirical relationship of magnitude and fault length. The results include many uncertainties. Maybe a large earthquake can rupture several branches. Also, the earthquakes on a fault have a similar depth. Only using the fault length to estimate the depth is not suitable.

We acknowledge that a large earthquake can, or indeed are likely to, rupture several fault branches, and thus our magnitude estimates should be considered maximum estimates. This is stated in the caption to Figure 8 *"Hillshade map of the Luangwa Rift showing estimated maximum earthquake magnitudes for each fault (orange circles) alongside previous recorded seismicity."* We also point out that multi-fault rupture is a possibility *"it is also possible that multi-fault rupture may occur, which would increase the potential magnitude of future events."* [Line 460]. We also point out that segmented ruptures are likely *"these large magnitude events likely occur infrequently as >100 km long normal faults in southern Africa are often segmented"* [Lines 445-446]

Unfortunately, we were unable to determine the level of segmentation to the faults in the Luangwa Rift because the SRTM data we used didn't have the resolution to do this (see discussion in the manuscript section 5.3). Thus, we are unable to make estimates of the magnitudes of smaller segmented ruptures. **To make this clearer, we will amend the text in the discussion section as follows:**

*"We estimate that the Molaza Fault is a seismic source capable of hosting an earthquake with a maximum magnitude of Mw 7.8"* [Lines 432-433]

*"…with the 207 km long Chipola fault capable of hosting up to a $M_w$ 8.1 earthquake"* [Lines 440-441]

*"To estimate the magnitude of smaller segmented ruptures, we attempted to identify fault segments for the four best exposed faults in the Luangwa Rift by systematically measuring along-strike fault scarp heights (Figure 10)"* [Lines 449-450]

The depth parameter we have estimated is based on empirical scaling relationships (Leonard 2010) for the purpose of estimating seismic source parameters. Using these scaling relationships ensure that our estimates of fault length, width, single event displacement, and earthquake magnitude are all self-consistent. We acknowledge that ideally fault depth would be inferred from evidence such as well-located earthquakes (e.g., Stevens et al 2021). Unfortunately, these data are not available in the Luangwa Rift, and in these cases it is appropriate for the purposes of seismogenic source modelling to use empirical scaling relationships instead (Williams et al 2021).

We also make it clear in section 3.2.1 that our estimates of these parameters are subjective and liable to change "*these estimates are often subjective and liable to change*" [line 173-174]. Furthermore, despite the uncertainties, using the empirical relationships of Leonard, 2010 in a logic tree approach is currently the most up-to-date and robust method to determine seismic source parameters where these can't be measured directly. It is used around the world in the state of the art seismic hazard analysis (e.g. Pace et al., 2016; SRL in Italy; Chartier et al., 2017; NHESS in Greece; Tromans et al., 2019; BEE in UK), and the Leonard empirical relationships are most appropriate for this region (Stirling et al., 2013; BSSA). However, we acknowledge that we could better outline the uncertainties and the fact that these depths represent the down-dip extent from the top of the rupture rather than the surface of the Earth. Therefore, we will change 'Maximum Rupture Depth' to 'Rupture Depth Extent' throughout the manuscript to reflect this. We will also add text to the methods outlining our definition of Rupture Depth Extent:

"*We also calculate the rupture depth extent (RDE), which we define as the maximum depth of the rupture can extend downwards, measured from the top of the rupture (ie it can occur anywhere in the crust).*" [Lines 178-179; Section 3.2.1]

4. No age, no rates. So your slip rates and associate recurrence intervals are not reliable. Your database of active faults is very nice. Maybe your analysis is over-interpret

Although calculating the age of an offset sedimentary feature is still the most commonly used method for estimating the slip rate of a fault, this is not always possible, and in the last 5 years, alternatives based on geodetic measurements have been proposed. For example, Dolan and Meade (2017, G-cubed; doi.org/10.1002/2017GC007014) present geodetically derived slip rates for three strike slip faults in different tectonic settings, and Evans (2018, BSSA; doi.org/10.1785/0120170159) analyses 33 previously published geodetic fault slip rate estimates from California.

The method that we used to generate first-order estimates of slip rates in the Luangwa Rift is based on the systems-based method outlined in Williams et al. (2021; Solid Earth; doi.org/10.5194/se-12-187-2021). This method uses the geodetically derived plate motion model and partitions this onto faults in a manner consistent with observations and modelling of the across-rift strain distribution in incipient amagmatic continental rifts (e.g., Muirhead et al 2019, Shillington et al 2020, Wedmore et al 2020).Thus, as the plate motion model produces an extension rate, we can determine an estimate for fault slip rate without requiring the age of any offset sediments.

This approach of combining geodetic and geologic observations to constrain fault slip rates in the absence of dated offset markers is entirely consistent with other regions around the world (e.g., Perea et al 2006 in Iberia; Dawson et al 2013 in California; and Seebeck et al 2022 in New Zeland). Furthermore, when measured independently from the 'systems-based' approach, and from offset dated seismic reflectors, the slip rate estimates of intrabasin faults in Lake Malawi are found to be within error of each other (Shillington et al 2020; Williams et al *in review*).

We are careful to apply this model within limitations, and we will update the text to state this:

"*We only apply the systems-based method to the two border faults, as these faults do not have any other faults across-strike for most of their length. Therefore, it is reasonable to assume as a first-order estimate that they take up all of the geodetic extension rate in this region. In addition, as the other faults in our database have a more complex across strike distribution, we do not attempt to determine a first-order estimate of their slip rates*" **[lines 191-195; section 3.2.2].**

5. In L274, The minimum heights using the SRTM data are 2-3 m, but your results show a high resolution like 0.08 m or 0.2 m. It is impossible.

**We will remove the figure quoted to two decimal places and thank the reviewer for pointing this out**. In all cases, we have only resolved scarp height values to one decimal place. As these are median values of many estimates of scarp height along strike, they are averages of many values, therefore it is appropriate to quote to 1 decimal place, as this is reasonably measurable given our Monte Carlo methodology (see response to R3). The minimum heights figure is a reference to the minimum scarp height that we can reliably confirm is caused by a fault offset given the quality of the SRTM data and the other topographic noise (e.g. vegetation, buildings etc) in the region, but does not refer to the precision of the measurement we can make. This was stated in the text as follows *"The minimum resolvable scarp heights that we are able to measure using the SRTM data is 2-3 m."*[lines 294-295]

6. L169-170: this sentence is not clear. Please check it.

**We will split this sentence into two sentences:**

"*However, these estimates are often subjective and more liable to change than the objective observational data stored in an active fault database. Therefore, we store these subjective estimates separately from the LRAFD, following other similar studies (e.g. Faure Walker et al., 2021; Williams et al., 2021).*

7. Line274: was—are

**Done**

8. Line 286: add m.

**Done**

9. Figure 4 show a nearly N-S-striking linear feature between faults 11 and 15 on the west side of the rift.

This feature did not meet our requirements to be identified as an active fault as no offset features were observed, no fault was identified in the geological map, and no other criteria were met to deem this an active fault. There are many linear-features within the study area, but not all are active faults as the large-scale metamorphic banding is often seen as in topographic maps of the region as a result of some foliation layers being less erodible. **We will add two sentences to the methods to make this clear and avoid any confusion:**

*"Care was taken to ensure that linear features that represent less erodible metamorphic bands within the foliated basement rock, which are often seen in southern Africa, were not incorrectly identified as active faults. By looking for similar topographic gradients either side of the fault, slope maps were particularly useful to ensure that on offset sedimentary landforms, the same geomorphic units were present on either side of the fault."* **[Line 119-122; section 3.1.1]**

10. Please show the profile locations of figure 9 in the other figure.

**We will add the coordinates of the profiles to Figure 9, as plotting the locations on other figures doesn't easily convey the location given the large scale regional maps that we've plotted.**

---

## Author Comment (AC2)

Dear Editor,

We thank the reviewers for a constructive set of reviews that will improve the manuscript. We have responded in detail to them in the attached document, and included text that we would include in the revised manuscript.

Thanks,

Luke Wedmore (on behalf of co-authors).

Our responses to the reviewer 2's comments are shown in blue text. **Specific changes that we will make to the text are in bold font.** The line numbers quoted refer to the revised version of the manuscript that we will resubmit if invited.

RC2

**Specific comments:**

Using an empirical relationship between the fault length and moment magnitude of earthquakes, the authors conclude that the fault could induce earthquakes up to Mw 8.1, a value that is greater than the historically recorded events in southern Africa. My concern is how confident are you with the resulting estimates, because it seems to me that there's a certain degree of uncertainty here.

We agree with the reviewer that there is a degree of uncertainty here, particularly because there is *a lot* of evidence from other rifts in East Africa that faults are highly segmented, and thus less likely to rupture along the whole extent. We attempted to look for evidence of segmentation along four of the faults (Chipola, Molaza, Chitumbi and Kabungo). Unfortunately, we concluded that the SRTM data that we used was not of sufficient resolution to allow us to resolve any segment boundaries. Nor is segmentation alone a preclude to these fault's hosting Mw 8.1, given that earthquakes can rupture through segment barriers (Hamling et al 2017, Du Ross et al 2016)

Furthermore, we outlined that smaller magnitude events are more likely that the headline Mw8.1 figure and outline a method to calculate the probability of smaller magnitude events: "*although we do not directly observe evidence for fault segmentation in the Luangwa Rift, the data provided here can be used to incorporate small ruptures along these faults into seismic hazard assessment by combining the provided slip rate, fault area, and magnitude estimates with a regional b-value (Poggi et al., 2017) and the methodology developed by Youngs and Coppersmith (1985) to develop continuous recurrence models for these sources (Williams et al 2022)*" [lines 456-459; Section 5.3]

**However, in response to the reviewers concerns and also reviewer 1, we will amend the text to make it clear that we consider these the maximum possible magnitude events, and to explicitly clarify that we attempted to assess segmentation:**

"***We estimate that the Molaza Fault is a seismic source capable of hosting an earthquake with a maximum magnitude of Mw 7.8***" [Lines 432-433]

*"…with the 207 km long Chipola fault capable of hosting up to a $M_w$ 8.1 earthquake"* [Lines 440-441]

*"To estimate the magnitude of smaller segmented ruptures, we attempted to identify fault segments for the four best exposed faults in the Luangwa Rift by systematically measuring along-strike fault scarp heights (Figure 10)"* [Lines 449-450]

**Technical corrections:**

- Line 57: 'Figure 1 & 2' should be 'Figures 1 & 2', and same for the rest, i.e., lines 216, 224, 225, 228;
- Line 105: The abbreviation was 'EARS' (Line 54), not 'EAR', same for Line 248;
- Lines 112 - 113: The same sentence appears as the first sentence of the last paragraph;
- Lines 120 and 123: The audience would be benefited with the full names of GEM (Global Earthquake Model) and GIS (Geographic Information System);
- Lines 168 - 170: Please check the sentence;
- Lines 186 and 187: Dav or dav?
- Line 213: Do you mean Figure 5?
- Line 261: trees?
- Line 394: The abbreviation of 'LRZ' has not been indicated before.

**We will correct all of these** – we thank the reviewer for they helpful comments and careful review.

---

## Author Comment (AC3)

Dear Editor,

We thank the reviewers for a constructive set of reviews that will improve the manuscript. We have responded in detail to them in the attached document, and included text that we would include in the revised manuscript.

Thanks,

Luke Wedmore (on behalf of co-authors).

Our responses to the reviewer 3's comments are shown in blue text. **Specific changes that we will make to the text are in bold font.** The line numbers quoted refer to the revised version of the manuscript that we will resubmit if invited.

RC3

You may consider also the work by Delvaux et al. (2012) on the segmentation and scarp height of the Kanda fault in the Rukwa rift.

- **Delvaux, D.**, Kervyn, F., Macheyeki, A.S., Temu, E.B. (2012). Geodynamic significance of the TRM segment in the East African Rift (W-Tanzania): active tectonics and paleostress in the Ufipa plateau and Rukwa basin. Journal of Structural Geology, 37, 161-180. DOI: 10.1016/j.jsg.2012.01.008.

In this work, we stress the importance of field measurement of the scarp height. We found that the local fault morphology might influence the calculated scrap height. This means that the selection of the site for performing the fault scarp profile is important and difficult to perform on the basis of remote sensing only. The question is thus how did you selected the sites for extracting scarp profiles and how did you took into consideration the local geomorphology associated to the fault?

We completely agree with the reviewer. Careful field observations to select a site to measure the scarp height is _always_ the best method to use. Such an approach was not possibly over the past few years due to covid, yet it was important to conduct this study as the area is rapidly developing so it is vital that constraints are placed on seismic hazard before this happens.

There are some benefits to the approach we have taken (ie using remote sensing only). Remote sensing allows a larger geographic area to be covered than field studies – it would be difficult, expensive, and very time consuming to measure the height of the ~200 km long Chipola fault using field studies alone. It also allows us to take measurements in areas that would be very difficult to access in the field.

**We will amend the text to make it clear that fieldwork is required to select the best sites for scarp height measurements and cite the paper suggested by the reviewer:**

*"Selecting sites for measuring displacement caused by earthquakes across faults is best performed by careful assessment of local geomorphology in the field (Delvaux et al. 2012, Bubeck et al. 2015). This was not possible in this study as the Covid19 pandemic restricted travel. Instead, we used SRTM data to measure the displacement across the faults, which enabled us to take multiple measurements across a large geographic area in a national park that is not always accessible for fieldwork."* [Line 215-219; Section 3.2.3]

As a consequence of using sub-optimal remote sensing methods, we implemented a range of technical measures to ensure our scarp height calculations are as accurate as possible:
- We selected the sites for extracting the scarp profiles on a systematic basis, sampling the digital elevation model every 30m, but then stacking these profiles together at intervals of 120 m.
- Individual profile samples were extracted perpendicular to the *local* strike of the fault.
- The purpose of the stacking is to remove the effects of localised changes in fault morphology, or other sources of random noise in the digital elevation model. However, this does not always successfully remove local variations in fault morphology, and thus when measuring the height of the stacked profiles, we are careful to only measure profiles that show a clear fault offset. For example, we would look for a clear scarp, and upper and lower surfaces of a similar angle, with no significant disturbances to the slope. Hence, although we were not able to use field observations to find the best sites to measure scarp heights, by instead using multiple scarp height measurements (and far more than could be practically measured in the field) to help smooth out random noise, we have in part negated this challenge.
- The code that we use to measure the height of the fault scarp in the profile (an adapted version of the code published in Hodge et al., 2019, Solid Earth) has additional quality checks to ensure the measured profile is not influenced by local morphological changes:
    o The code takes a Monte-Carlo approach to sampling the upper and lower slopes, and selects 10,000 random subsets of the points on each slope, so that any small-scale morphological disturbance shouldn't be reflected in the final measurement.
    o If more than one maximum in the gradient of the slope across the scarp itself is detected, then our code does not make a measurement in that location.

This approach has been successfully implemented, and verified with field measurements, in southern Malawi (e.g. Wedmore et al., 2020, Tectonics). Thus, we are confident that it has been successful in Zambia, despite not being able to do the verification in the field.

**We will amend the text to the following text to make this clear and add the reference suggestion from the review as follows:**

*"We extracted topographic profiles every 30 m oriented perpendicular the local strike of the fault. As local fault morphology can influence the calculated scarp height (Delvaux et al., 2012) we stacked the profiles at 120 m intervals along strike to filter short-wavelength topographic features such as vegetation or human structures that are unrelated to active faulting. Profiles were then visually inspected, for a clear scarp, with footwall and hanging wall slopes above and below the scarp of approximately the same angle. The scarp height of profiles that passed our visual inspection were measured following the approach of*

*Wedmore et al. (2020b), which includes inbuilt Monte Carlo sampling of random subsets of the footwall and hanging wall slopes to prevent small scale local morphological disturbances affecting the calculated scarp height. Verification with detailed field measurements was not possible during this study because of travel restrictions during the Covid19 pandemic, but these methods have previously been used and verified in southern Malawi, which has a similar tectonic setting and terrain (Wedmore et al., 2020a, b)"* [lines 219-229; Section 3.2.3]